# Cholesterol-Lowering Gene Therapy Prevents Heart Failure with Preserved Ejection Fraction in Obese Type 2 Diabetic Mice

**DOI:** 10.3390/ijms20092222

**Published:** 2019-05-06

**Authors:** Joseph Pierre Aboumsallem, Ilayaraja Muthuramu, Mudit Mishra, Bart De Geest

**Affiliations:** Centre for Molecular and Vascular Biology, Department of Cardiovascular Sciences, Catholic University of Leuven, 3000 Leuven, Belgium; josephpierre.aboumsallem@kuleuven.be (J.P.A.); illas1985@gmail.com (I.M.); mudit.mishra@kuleuven.be (M.M.)

**Keywords:** gene therapy, heart failure with preserved ejection fraction, type 2 diabetes, oxidative stress, hypercholesterolemia

## Abstract

Hypercholesterolemia may be causally related to heart failure with preserved ejection fraction (HFpEF). We aimed to establish a HFpEF model associated with hypercholesterolemia and type 2 diabetes mellitus by feeding a high-sucrose/high-fat (HSHF) diet to C57BL/6J low-density lipoprotein receptor (LDLr)^−/−^ mice. Secondly, we evaluated whether cholesterol-lowering adeno-associated viral serotype 8 (AAV8)-mediated LDLr gene transfer prevents HFpEF. AAV8-LDLr gene transfer strongly (*p* < 0.001) decreased plasma cholesterol in standard chow (SC) mice (66.8 ± 2.5 mg/dl versus 213 ± 12 mg/dl) and in HSHF mice (84.6 ± 4.4 mg/dl versus 464 ± 25 mg/dl). The HSHF diet induced cardiac hypertrophy and pathological remodeling, which were potently counteracted by AAV8-LDLr gene transfer. Wet lung weight was 19.0% (*p* < 0.001) higher in AAV8-null HSHF mice than in AAV8-null SC mice, whereas lung weight was normal in AAV8-LDLr HSHF mice. Pressure–volume loop analysis was consistent with HFpEF in AAV8-null HSHF mice and showed a completely normal cardiac function in AAV8-LDLr HSHF mice. Treadmill exercise testing demonstrated reduced exercise capacity in AAV8-null HSHF mice but a normal capacity in AAV8-LDLr HSHF mice. Reduced oxidative stress and decreased levels of tumor necrosis factor-α may mediate the beneficial effects of cholesterol lowering. In conclusion, AAV8-LDLr gene therapy prevents HFpEF.

## 1. Introduction

Overwhelming epidemiological evidence shows that diabetes mellitus is independently associated with heart failure incidence, with the risk being increased by more than twofold in men and by more than fivefold in women [1]. The most prevalent phenotype of heart failure in patients with obesity and diabetes mellitus is heart failure with preserved ejection fraction (HFpEF) (ejection fraction ≥50%) [2]. Inflammation has been proposed to be the primary pathophysiological driver of HFpEF, whereas cardiomyocyte loss and stretch are the principal drivers of heart failure with reduced ejection fraction (HFrEF) (ejection fraction ≤40%) [3,4]. Chronic inflammation in adipose tissue is a hallmark of obesity. Systemic effects of adipose tissue inflammation or local inflammation in the epicardial fat may have detrimental effects on the myocardial structure and function [5]. Capillary rarefaction and impaired myocardial microcirculation, increased ventricular stiffness due to myocardial fibrosis, and abnormal isovolumetric relaxation are typical features associated with HFpEF that may be potentiated by local and systemic inflammation [6].

Diabetes mellitus is associated with dyslipidemia, which may also result in pro-inflammatory, pro-oxidative, and pro-fibrotic effects contributing to HFpEF. Dyslipidemia increases the incidence rate of heart failure that is independent of its association with myocardial infarction [7]. In the prospective Swedish Heart Failure Registry, the use of 3-hydroxy-3-methylglutaryl-coenzyme A reductase inhibitors (statins) was associated with improved outcomes in patients with HFpEF [8]. Taken together, hypercholesterolemia may be causally related to HFpEF even in the absence of coronary artery disease, but this has not been demonstrated until now.

Given the profound difference in the metabolism of fructose and glucose [9], the hypothesis has been put forward that fructose may have a unique and distinct contributing role in the aetiology of obesity, type 2 diabetes mellitus, and cardiovascular disease [10,11]. The major source of fructose in the diet are added sugars that come under the form of sucrose or of high-fructose corn syrup used in the production of sugar-sweetened beverages [10,11]. Since obesity and type 2 diabetes mellitus are to a major extent related to environmental factors, our aim was to establish a model of diabetic HFpEF induced by a high-sugar/high-fat (HSHF) diet in C57BL/6J low density receptor (LDLr) deficient mice. Secondly, we intended to test the hypothesis that adeno-associated viral (AAV) serotype 8-mediated cholesterol-lowering gene therapy prevents HFpEF in this model.

## 2. Results

### 2.1. The HSHF Diet Induces Obesity and Type 2 Diabetes Mellitus in Female C57BL/6J LDLr^−/−^ Mice

The global study design evaluating the effect of AAV8-LDLr gene transfer on the development of HFpEF in HSHF diet mice is illustrated in Figure 1. Gene transfer in female C57BL/6J LDLr^−/−^ mice with the control vector AAV8-null or with AAV8-LDLr containing hepatocyte-specific transcriptional regulatory sequences to induce expression of the murine LDLr in parenchymal liver cells was performed at the age of 11 weeks. The high-sugar/high-fat (HSHF) diet was initiated in two groups one week later. The two other groups received a standard chow (SC) diet. Endpoint analyses in all groups were performed at the age of 28 weeks.

Murine LDLr expression in the liver of C57BL/6 LDLr^−/−^ mice, wild-type C57BL/6J mice, and C57BL/6J LDLr^−/−^ mice treated with 3 × 10^12^ genome copies/kg of AAV8-LDLr is illustrated in Figure 2. Murine LDLr was undetectable in C57BL/6J LDLr^−/−^ mice. Murine LDLr expression 17 weeks after gene transfer with AAV8-LDLr in C57BL/6J LDLr^−/−^ mice was 4.26-fold (*p* < 0.05) higher compared to wild-type C57BL/6J mice.

The time course of the body weight in SC-diet and HSHF-diet mice is shown in Figure 3A. Compared to AAV8-null SC diet mice, the body weight in AAV8-null HSHF diet mice was 1.16-fold (*p* < 0.0001) higher at 4 weeks, 1.28-fold (*p* < 0.0001) higher at 8 weeks, 1.38-fold (*p* < 0.0001) higher at 12 weeks, and 1.49-fold (*p* < 0.0001) higher at 16 weeks. AAV8-LDLr gene transfer did not affect body weight in SC diet mice but attenuated weight gain in HSHF diet mice. Body weight in AAV8-LDLr HSHF diet mice was reduced by 10.7% (*p* < 0.001) at 4 weeks, by 15.3% (*p* < 0.0001) at 8 weeks, by 16.6% (*p* < 0.0001) at 12 weeks, and by 18.7% (*p* < 0.0001) at 16 weeks compared to AAV8-null HSHF diet mice. Daily food intake measured during the entire duration of the experiment was not significantly different between AAV8-null HSHF diet mice (5.93 ± 0.37 g/mouse/day) and AAV8-LDLr HSHF diet mice (6.23 ± 0.33 g/mouse/day). The HSHF diet induced diabetes mellitus (Figure 3B). Blood glucose levels in AAV8-null HSHF diet mice were 1.26-fold (*p* < 0.0001) higher at 8 weeks and 1.26-fold (*p* < 0.0001) higher at 16 weeks compared to AAV8-null SC diet mice. Blood glucose levels in AAV8-LDLr HSHF diet mice were significantly lower at 8 weeks (*p* < 0.05), at 12 weeks (*p* < 0.05), and at 16 weeks (*p* < 0.0001) compared to AAV8-null HSHF diet mice.

To evaluate the effect of AAV8-LDLr gene transfer on adiposity induced by the HSHF diet, a histological analysis of the gonadal fat pad was performed (Figure 4). The adipocyte cross-sectional area was 2.52-fold (*p* < 0.001) higher in AAV8-null HSHF diet mice than in AAV8-null SC diet mice. AAV8-LDLr gene transfer abrogated adipocyte hypertrophy induced by the HSHF diet (Figure 4A). The adipocyte cross-sectional area was reduced by 68.4% (*p* < 0.001) in AAV8-LDLr HSHF diet mice compared to AAV8-null HSHF diet mice. Adipocyte density was reduced by 59.5% (*p* < 0.001) in AAV8-null HSHF diet mice compared to AAV8-null SC diet mice and was not reduced at all in AAV8-LDLr HSHF diet mice.

### 2.2. Effect of AAV8-LDLr Gene Transfer on Metabolic Parameters in SC diet and in HSHF Diet Mice

The HSHF diet induced hyperinsulinemia. Plasma insulin levels were 4.93-fold (*p* < 0.001) and 3.98-fold (*p* < 0.001) higher in AAV8-null HSHF diet mice and AAV8-LDLr HSHF diet mice, respectively, compared to respective SC diet groups (Figure 5A). Plasma free fatty acids (FFA) were 27.2% (*p* < 0.01) lower in AAV8-LDLr HSHF diet mice than in AAV8-null HSHF diet mice (Figure 5B). No significant differences of plasma adiponectin levels were observed (Figure 5C).

Lipid levels in SC diet mice and in HSHF diet mice are represented in Figure 6. Plasma cholesterol was 2.18-fold (*p* < 0.001) higher in AAV8-null HSHF diet mice compared to AAV8-null SC diet mice (Figure 6A). AAV8-LDLr gene transfer decreased plasma cholesterol by 68.6% (*p* < 0.001) and by 81.8% (*p* < 0.001) in SC diet mice and in HSHF diet mice, respectively. Plasma non-HDL cholesterol levels were increased by 2.32-fold (*p* < 0.001) in AAV8-null HSHF diet mice compared to AAV8-null SC diet mice (Figure 6B). Plasma non-HDL cholesterol levels were 77.8% (*p* < 0.001) and 89.8% (*p* < 0.001) lower in AAV8-LDLr SC diet mice and AAV8-LDLr HSHF diet mice, respectively, compared to respective AAV8-null groups. Plasma HDL cholesterol was 19.1% (*p* < 0.01) lower in AAV8-LDLr HSHF diet mice than in AAV8-null HSHF diet mice (Figure 6C). Plasma triglyceride levels were not significantly different between different groups (Figure 6D).

### 2.3. AAV8-Gene Transfer Potently Counteracts Cardiac Hypertrophy and Pathological Remodeling in Female C57BL/6J LDLr^−/−^ Mice Fed the HSHF Diet

Organ and tissue weights in AAV8-null SC diet mice, AAV8-LDLr SC diet mice, AAV8-null HSHF diet mice, and AAV8-LDLr HSHF diet mice are represented in Table 1. The HSHF diet induced cardiac hypertrophy in AAV8-null-treated mice as evidenced by a 1.19-fold (*p* < 0.001) increase of the wet heart weight and a 1.18-fold increase (*p* < 0.001) increase of heart weight normalized to tibia length. Wet heart weight was 16.0% (*p* < 0.001) lower in AAV8-LDLr HSHF diet mice compared to AAV8-null HSHF diet mice. Left ventricular weight and right ventricular weight in AAV8-null HSHF diet mice were 1.23-fold (*p* < 0.001) and 1.35-fold (*p* < 0.01) higher, respectively, than in AAV8-null SC diet mice (Table 1). Left ventricular weight and right ventricular weight in AAV8-LDLr HSHF diet mice were reduced by 16.7% (*p* < 0.001) and by 24.1% (*p* < 0.01), respectively, compared to AAV8-null HSHF diet mice. Wet lung weight was increased 1.19-fold (*p* < 0.001) in AAV8-null HSHF diet mice compared to AAV8-null SC diet mice indicating the presence of heart failure. In contrast, wet lung weight was not higher in AAV8-LDLr HSHF diet mice than in AAV8-LDLr SC diet mice and was 12.6% (*p* < 0.001) lower than in AAV8-null HSHF diet mice. Moreover, liver weight, kidney weight, and spleen weight were significantly (*p* < 0.001) increased in AAV8-null HSHF diet mice compared to AAV8-null SC diet mice, whereas no increase was observed in AAV8-LDLr HSHF diet mice. The degree of liver steatosis was 1.98-fold (*p* < 0.01) higher in AAV8-null HSHF diet mice than in AAV8-null SC diet mice (Figure 7A). Liver steatosis was reduced by 38.1% (*p* < 0.01) in AAV8-LDLr HSHF diet mice compared to AAV8-null HSHF diet mice. Representative photomicrographs illustrating hematoxylin- and eosin-stained liver tissue are shown in Figure 7B. Taken together, the HSHF diet caused cardiac hypertrophy and heart failure, which were potently counteracted by AAV8-LDLr gene transfer.

Morphometric analysis demonstrated that left ventricular wall area and anterior wall thickness in AAV8-null HSHF diet mice were 1.24-fold (*p* < 0.001) and 1.27-fold (*p* < 0.001) larger, respectively, than in AAV8-null SC diet mice (Table 2). Left ventricular wall area and anterior wall thickness in AAV8-LDLr HSHF diet mice were decreased by 13.8% (*p* < 0.01) and by 15.6% (*p* < 0.001), respectively, compared to AAV8-null HSHF diet mice. This was paralleled at the microscopic level by a significant (*p* < 0.001) increase of the cardiomyocyte cross-sectional area in AAV8-null HSHF diet mice compared to AAV8-null SC diet mice and a significantly (*p* < 0001) lower cardiomyocyte cross-sectional area in AAV8-LDLr HSHF diet mice than in AAV8-null HSHF diet mice. Moreover, the hypertrophy in HSHF diet mice was pathological as evidenced by the decrease of capillary density (*p* < 0.01) and relative vascularity (*p* < 0.01) and by the increased myocardial fibrosis (*p* < 0.001). Capillary density and relative vascularity were prominently (*p* < 0.01) increased in AAV8-LDLr HSHF diet mice compared to AAV8-null HSHF diet mice. Moreover, the degree of interstitial fibrosis and of perivascular fibrosis in AAV8-LDLr HSHD diet mice were 37.6% (*p* < 0.001) and 22.0% (*p* < 0.05) lower, respectively, in AAV8-LDLr HSHF diet than in AAV8-null HSHF diet mice (*p* < 0.001). Representative Sirius Red-stained cross-sections of SC diet and HSHF diet mice are shown in Figure 8. Laminin-stained cardiomyocytes, CD31-positive capillaries, Sirius Red-stained collagen viewed under polarized light, and immunosections stained for 3-nitrotyrosine in all four groups are illustrated by the representative photomicrographs in Figure 9. In aggregate, the HSHF diet leads to pathological remodeling, which is efficaciously antagonized by AAV8-LDLr gene transfer.

### 2.4. Cardiac Dysfunction in C57BL/6J LDLr^−/−^ Mice Fed the HSHF Diet Is Consistent with HFpEF, whereas AAV8-LDLr Gene Transfer Completely Normalizes Cardiac Function

Pressure–volume loop data obtained in AAV8-null SC diet mice, AAV8-LDLr SC diet mice, AAV8-null HSHF diet mice, and AAV8-LDLr HSHF diet mice are summarized in Table 3. The HSHF diet induced pronounced abnormalities of systolic function and of diastolic function in AAV8-null-treated mice. Systolic dysfunction in AAV8-null HSHF diet mice was evidenced by a 28.9% (*p* < 0.001) decrease in the maximal rate of isovolumetric contraction (dP/dt_max_), a 19.7% (*p* < 0.05) reduction of the preload recruitable stroke work (PRSW), a 29.7% (*p* < 0.05) decrease of end-systolic elastance (E_es_), and a 25.7% reduction (*p* < 0.01) of the absolute value of the peak emptying rate (dV/dt_max_) compared to AAV8-null SC diet mice. AAV8-null HSHF diet mice were characterized by a 20.1% (*p* < 0.01) reduction of the absolute value of the maximal rate of isovolumetric relaxation (dP/dt_min_), a 1.27-fold (*p* < 0.001) increase in the time constant of isovolumetric relaxation tau, a 1.78-fold (*p* < 0.01) increase in the slope of the end-diastolic pressure–volume relationship (EDPVR), and a 28.5% (*p* < 0.01) reduction of the peak filling rate (dV/dt_min_). These alterations led to a 27.9% (*p* < 0.001) and a 28.8% (*p* < 0.001) reduction of stroke volume and of cardiac output, respectively, in AAV8-null HSHF diet mice compared to AAV8-null SC diet mice. Ejection fraction was above 50% in AAV8-null HSHF diet mice, indicating that these mice were affected by HFpEF. Consistently, the end-diastolic volume (EDV) was reduced by 20.3% (*p* < 0.05) in AAV8-null HSHF diet mice compared to AAV8-null SC diet mice. Ventriculo–arterial coupling was impaired in AAV8-null HSHF diet mice, as evidenced by a 2.07-fold (*p* < 0.05) increase in the ventriculo–arterial coupling ratio. AAV8-LDLr gene transfer induced a supernormal cardiac function in SC diet mice, as evidenced by a 19.3% (*p* < 0.01) increase in dP/dt_max_, a significantly increased ejection fraction (*p* < 0.01), and a 1.26-fold (*p* < 0.05) increase in the absolute value of the peak emptying rate (dV/dt_max_). More importantly, AAV8-LDLr gene transfer in HSHF diet mice normalized systolic function (dP/dt_max_, PRSW, E_es_) and diastolic function (dP/dt_min_, tau, slope EDPVR). Moreover, stroke volume, cardiac output, and ventriculo–arterial coupling were completely normalized. Taken together, cholesterol-lowering gene therapy completely prevents diabetic HFpEF in C57BL/6J LDLr^−/−^ mice fed the HSHF diet.

### 2.5. Exercise Capacity Is Severely Limited in C57BL/6J LDLr^−/−^ Mice Fed the HSHF Diet and Is Completely Normalized by AAV8-LDLr

Exercise treadmill testing was applied to evaluate exercise capacity in the two SC diet groups and in the two HSHF diet groups (Figure 10A). The distance covered was reduced by 62.0% (*p* < 0.001) in AAV8-null HSHF diet compared to AAV8-null SC diet mice (Figure 10B). In contrast, the exercise capacity was completely normal in AAV8-LDLr HSHF diet, and the distance covered was 2.60-fold (*p* < 0.001) higher than in AAV8-null HSHF diet mice. Lactate post-exercise levels were 1.60-fold (*p* < 0.01) higher in AAV8-null HSHF diet than in AAV8-null SC diet mice (Figure 10C). A 46.9% (*p* < 0.001) reduction of lactate post-exercise levels was observed in AAV8-LDLr HSHF diet mice compared to AAV8-null HSHF diet mice.

### 2.6. Pronounced Increase of Tumor Necrosis Factor (TNF)-α Levels in C57BL/6J LDLr^−/−^ Mice Fed the HSHF Diet Is Completely Abrogated by AAV8-LDLr Gene Transfer

TNF-α may play a role in the progression of heart failure. Plasma levels of TNF-α were 7.38-fold (*p* < 0.001) higher in AAV8-null HSHF diet than in AAV8-null SC diet mice (Figure 11). An 82.7% (*p* < 0.001) reduction of plasma TNF-α was observed in AAV8-LDLr HSHF diet mice compared to AAV8-null HSHF diet mice (Figure 11).

## 3. Discussion

The principal findings of the current study are that 1) feeding an HSHF diet in female C57BL/6J LDLr^−/−^ mice induces obesity, hyperinsulinemia, type 2 diabetes mellitus, and HFpEF; 2) cholesterol-lowering gene therapy potently counteracted structural remodeling and normalized cardiac function in HSHF diet mice; 3) HFpEF in HSHF diet mice was completely prevented by cholesterol-lowering gene therapy, as evidenced by the normal wet lung weight and by the normal exercise capacity; 4) cholesterol-lowering gene therapy strongly reduced nitro-oxidative stress and plasma TNF-α levels in HSHF diet mice, which may be important mediators of the beneficial effects of this treatment on cardiac structure and function.

Previously, we demonstrated that cholesterol-lowering gene therapy improves survival, counteracts left ventricular hypertrophy, attenuates metabolic remodeling, and improves cardiac function in a HFrEF model [12]. In this model, pressure overload-induced cardiomyopathy was induced by transverse aortic constriction. In contrast to the previous study, the current investigation deals with a model of diabetic HFpEF. In contrast to HFrEF, no treatment options are at present available for HFpEF, and this condition represents an unmet therapeutic need. The presence of heart failure in this model is evidenced by the increased wet lung weight and the strongly reduced exercise capacity in AAV8-null HSHF diet mice. The profound impact of AAV8-LDLr gene therapy on cardiac function evaluated by pressure–volume loop measurements is corroborated by the normalization of the exercise capacity in AAV8-LDLr HSHF diet mice. Moreover, pressure–volume loop analysis indicated that several hemodynamic parameters were ameliorated in AAV8-LDLr SC diet mice compared to AAV8-null SC diet mice. Therefore, AAV8-LDLr gene transfer also improved cardiac function in the absence of type 2 diabetes mellitus.

Oxidative stress, which reflects an excess production of reactive oxygen species relative to antioxidant defense systems, contributes to cardiac remodeling and the development of heart failure [13]. Mitochondria constitute the principal intracellular site of reactive oxygen species generation [14]. Mitochondria from hypercholesterolemic LDLr^−/−^ mice are characterized by a preserved oxidative phosphorylation efficiency but have at the same time a higher net production of reactive oxygen species [15]. When the capacity of cellular antioxidant defenses is insufficient, reactive oxygen species react with cellular lipids, proteins, and DNA, causing mitochondrial [14] and cellular damage [16,17,18,19,20,21]. Key processes underlying diabetic cardiac remodeling, namely inflammation, cardiomyocyte hypertrophy, and apoptosis, and myocardial fibrosis are all redox sensitive [22]. The reduced nitro-oxidative stress following AAV8-LDLr gene therapy may be an important contributor to the prevention of cardiac remodeling and HFpEF in HSHF diet mice.

Evidence for a causal role of TNF-α in diabetic heart failure has been provided in a rat model of streptozotocin-induced diabetes mellitus [23]. Elevated inflammatory cytokines are a biomarker of chronic heart failure, but whether inflammation plays a causal role in disease progression has not been established in humans [24]. The effect of cytokines should not be considered in isolation but is dependent on the global biological and inflammatory context [25]. Therefore, decreased TNF-α levels following AAV8-LDLr gene transfer should be considered in the global context of biological alterations induced by cholesterol-lowering gene therapy.

The HSHF diet resulted in marked hyperglycemia and marked hyperinsulinemia, indicating that insulin resistance and type 2 diabetes mellitus were induced by this diet. A limitation of the current study is that the weights of fat pads were not directly determined. However, the quantitative data on adipocyte hypertrophy provide evidence for the presence of an enlargement of adipose tissue [26]. Type 2 diabetes mellitus is a systemic disease that affects several organs. Therefore, multiple pathologies may cluster within one and the same patient: HFpEF, diabetic nephropathy, and non-alcoholic fatty liver disease (NAFLD) [27]. The HSHF diet model was characterized by the presence of hepatic steatosis, which is concordant with earlier observations [28]. NAFLD is also highly prevalent in subjects with obesity and type 2 diabetes mellitus [29]. The global prevalence of NAFLD in the world is as high as one billion [30]. In patients with type 2 diabetes mellitus without known history of cardiac disease, NAFLD was associated with early features of left ventricular diastolic dysfunction [31].

A surprising finding in this study is that AAV8-LDLr gene therapy attenuated the weight gain in mice receiving the HSHF diet. This difference is not explained by a difference in food intake since daily food intake was not different between AAV8-null HSHF diet mice and AAV8-LDLr HSHF diet mice. In general, lipoprotein metabolic pathways are a pivotal contributor to the development of obesity. Modulation of lipolysis, receptor-mediated clearance of triglyceride-rich lipoproteins, and cross-talk between the liver and adipose tissue are major regulators of energy expenditure, whole-body homeostasis, and body weight. Apolipoprotein E is a key protein in the clearance of chylomicron remnants, LDL, and very low-density lipoproteins (VLDL). Apolipoprotein E-deficient mice are resistant to diet-induced obesity, insulin resistance, and glucose intolerance [32,33,34]. Adipocyte-specific inactivation of the multifunctional receptor LDL receptor-related protein-1 (LRP1) resulted in delayed postprandial lipid clearance, reduced body weight, smaller fat stores, resistance to dietary fat-induced obesity, improved glucose tolerance, and elevated energy expenditure due to enhanced muscle thermogenesis [35]. The LDLr is also expressed in adipocytes [36]. Uptake and degradation of VLDL, intermediate density lipoprotein (IDL), and LDL particles may subsequently lead to the release of molecules with signaling properties [37,38]. Binding of apoB100-containing LDL to adipocytes via the LDL receptor inhibits intracellular noradrenaline-induced lipolysis in adipocytes [39].

Modulation of body weight by the LDLr may be tissue-dependent. On the one hand, absence of the LDLr in adipocytes of C57BL/6J LDLr^−/−^ mice may attenuate the increase in adiposity when placed on an HSHF diet. Indeed, body weight in AAV8-null C57BL/6J LDLr^−/−^ HSHF diet mice was 9.38% (*p* < 0.05) lower than in wild-type C57BL/6J HSHF diet mice with exactly the same genetic background (unpublished data). This difference is also consistent with observations in humans showing that body mass index in patients with heterozygous familial hypercholesterolemia is significantly lower compared to unaffected relatives [40]. On the other hand, the supraphysiological expression level of the murine LDLr in parenchymal liver cells of AAV8-LDL C57BL/6J LDLr^−/−^ HSHF diet mice may further attenuate weight gain by potentiation of postprandial catabolism of triglyceride-rich lipoproteins and may explain the significantly lower body weight and the absence of adipocyte hypertrophy in AAV8-LDL C57BL/6J LDLr^−/−^ HSHF diet mice compared to AAV8-null C57BL/6J LDLr^−/−^ HSHF diet mice.

Several lines of evidence indicate the existence of a link between cholesterol levels and diabetes risk. Major randomized clinical trials and meta-analyses of these trials have indicated that statin therapy, which inhibits 3-hydroxy-3-methylglutaryl-coenzyme A reductase, is associated with a modestly higher risk of developing type 2 diabetes mellitus in a dose-dependent fashion [41,42]. Moreover, Mendelian randomization studies have linked genetically-determined lower LDL cholesterol to increased risk of type 2 diabetes [43]. Subjects with lower circulating LDL cholesterol due to variants in the activity of 3-hydroxy-3-methylglutaryl-CoA reductase were characterized by higher glucose levels, increased weight, and elevated risk for type 2 diabetes mellitus [43]. Patients with heterozygous familial hypercholesterolemia appear to be protected against the development of type 2 diabetes mellitus [40]. A dose-dependent association was observed, with the lowest prevalence in subjects with the most severe phenotype in terms of plasma LDL cholesterol [40]. In contrast, AAV8-LDLr gene therapy had no impact on glucose levels and on plasma insulin levels in this study in mice fed the SC diet. On the other hand, glucose levels were lower in AAV8-LDLr HSHF diet C57BL/6J LDLr^−/−^ mice compared to AAV8-null C57BL/6J LDLr^−/−^ HSHF diet mice, but this observation is confounded by the difference in body weight of both conditions.

The impact of cholesterol-lowering therapies on clinical heart failure is an area full of controversy. The 3-hydroxy-3-methylglutaryl-coenzyme A reductase inhibitors (statins), which constitute the major class of hypolipidemic drugs, reduce heart failure events in primary and secondary prevention trials [44]. Moreover, the impact of statins was similar in subjects with a preceding myocardial infarction and in individuals without a prior myocardial infarction [44]. With longer follow-up, the preventive effect of statins on heart failure incidence may be more pronounced. During an extended follow-up of 15 years in the West of Scotland Coronary Prevention Study, hospitalization for heart failure was decreased by 43% in subjects receiving pravastatin compared with placebo recipients, whereas no effect on heart failure was observed in the first 5 years [45]. In clear contrast, statins do not appear to result in major beneficial effects in patients with established heart failure as has been demonstrated in the CORONA and GISSI-HF trials [46,47]. A general assumption is that the pathophysiological mechanisms that lead to the onset of heart failure are similar to those that underlie progression of this disorder [2]. One potential explanation for the differential effects of statins on prevention and treatment of heart failure is that this assumption is simply not consistently true. Another explanation is that beneficial effects of statins in patients with established disease are offset by untoward effects of statins, which are only unmasked once heart failure has developed. Statins have pleiotropic effects that are independent of cholesterol lowering. One of these cholesterol lowering-independent effects of statin lowering is the inhibition of endogenous coenzyme Q10 synthesis [48]. Coenzyme Q10 is a component of the electron transport chain and participates in aerobic cellular respiration in mitochondria. Low plasma levels of coenzyme Q10 are a risk factor for adverse outcomes in heart failure patients [49]. In contrast to statins and other cholesterol-lowering strategies, AAV8-LDLr gene transfer selectively decreases plasma cholesterol, since all its effects are mediated via increased LDLr expression in parenchymal liver cells. It remains to be established in an intervention study that AAV8-LDLr gene therapy can reverse established heart failure in this model of HSHF diet-induced HFpEF.

In conclusion, cholesterol-lowering gene therapy prevents cardiac remodeling, cardiac dysfunction, and HFpEF in a model of obesity and type 2 diabetes mellitus induced by an HSHF diet. This study corroborates the view that hypercholesterolemia may causally contribute to HFpEF.

## 4. Materials and Methods

### 4.1. Gene Therapy

Cholesterol-lowering gene therapy was performed using an adeno-associated viral (AAV) serotype 8 vector containing a hepatocyte-specific expression cassette to induce expression of the murine LDLr (AAV8-LDLr) [12,50]. The expression cassette of this vector consists of the 1272 bp DC172 promoter, comprising an 890 bp α_1_-antitrypsin promoter fused together with 2 copies of the 160 bp α_1_-microglobulin enhancer [50], upstream of the human A-I 5′UTR containing the first intron (247 bp) followed by the murine LDLr cDNA sequence (2598 bp), and the rabbit ß-globin polyadenylation signal (127 bp). Gene transfer was performed at the age of 11 weeks by tail vein injection of 3 x 10^12^ genome copies/kg of AAV8-LDLr. Control mice were treated with an equivalent dose of AAV8-null. The AAV8-null control vector contains the same transcriptional regulatory sequences compared to AAV8-LDLr but no insert. AAV vector production was performed as described [51].

### 4.2. In Vivo Experiments and Study Design

All experimental procedures in animals were executed in accordance with protocols approved by the Institutional Animal Care and Research Advisory Committee of the Catholic University of Leuven (Approval number: P191/2015, approval date: 5 November 2015) [52]. C57BL/6J LDLr^−/−^ mice, originally purchased from Jackson Laboratories (Bar Harbor, ME, USA), were locally bred at the semi-specific pathogen free facility of the Catholic University of Leuven at Gasthuisberg. The study design is illustrated in Figure 1. All experimental mice were female and were fed a standard chow (SC) diet (Ssniff Spezialdiäten GmbH, Soest, Germany) or were fed the pellet form of TestDiet 58Y1/5APC (London, WC1N3XX, England, United Kingdom) starting at the age of 12 weeks. This experimental high-sugar/high-fat (HSHF) diet was maintained for 16 weeks. The composition of TestDiet 58Y1/5APC (HSHF diet) has been described before [53]. Metabolizable energy from the standard chow is 13.5 MJ/kg (9 kJ% fat, 24 kJ% protein, 67 kJ% carbohydrates), whereas metabolizable energy from the HSHF diet is 19.5 MJ/kg (46.4 kJ% fat, 17.6 kJ% protein, 36.0 kJ% carbohydrates). Mono- and disaccharides in the HSHF diet consisted of high-fructose corn syrup-55 (17.5%) and sucrose (17.5%) [53]. All four groups of mice (AAV8-null SC diet, AAV8-LDLr SC diet, AAV8-null HSHF diet, AAV8-LDLr HSHF diet) were analyzed at the age of 28 weeks. In the first experimental layer, mice were assigned for hemodynamic quantification and histochemical and immunohistochemical analysis. The second experimental layer consisted of mice that did not undergo perfusion fixation and that were used for the quantification of tissue and organ weights.

Group assignment at the start of the study was performed randomly. No mice died during the study. No mice were excluded from the analysis. Endpoint analyses were performed by investigators who were blinded to the group allocation of the mice. Unblinding of animal numbers corresponding to specific allocation groups was performed at the completion of measurements.

### 4.3. Quantification of Murine LDLr Expression in the Liver by Western Blot

Liver tissue samples were isolated and immediately frozen in liquid nitrogen and stored at –80 °C. The extraction, blotting, and protein quantification procedures have been described in detail before [12]. The primary antibody for detection of the murine LDLr was obtained from Abcam (Cambridge, United Kingdom). The primary antibody for the quantification of β-tubulin was purchased from Cell Signaling Technology (Beverly, MA, USA). Protein expression was detected with SuperSignal West Pico chemiluminescence reagents (Thermo Fisher Scientific, Rockford, IL, USA) and quantified using Image Lab software (Bio-Rad Laboratories, Hercules, CA, USA). All protein levels were normalized to the β-tubulin protein level.

### 4.4. In Vivo Hemodynamic Pressure–Volume Loop Measurements

Invasive hemodynamic measurements were performed before sacrifice following anesthesia induced by intraperitoneal administration of 1.2 g/kg urethane (Sigma) [52]. Measurements were performed using Millar’s Mikro-Tip^®^ ultra-miniature pressure-volume (PV) loop catheter PVR-1035 (1.0 French polyimide catheter), the MPVS Ultra Single Segment Pressure-Volume Unit, and a PowerLab 16/35 data acquisition system (ADInstruments Ltd., Oxford, United Kingdom).

### 4.5. Quantification of Plasma Lipid Levels and Lipoprotein Cholesterol

Blood was obtained by puncture of the retro-orbital plexus. Anticoagulation was performed with 0.1 volume of 136 mmol/L trisodium citrate. Subsequently, plasma was immediately isolated by centrifugation at 1100× *g* for 10 min and stored at –20 °C [54,55,56].

Plasma cholesterol and lipoprotein cholesterol levels in C57BL/6J LDLr^−/−^ mice were determined using the Cholesterol Quantification kit from Sigma-Aldrich (Sigma, St. Louis, MO, USA). HDL and non-HDL lipoproteins were separated by ultracentrifugation as described [12]. Plasma triglyceride concentration was quantified using the Triglyceride Quantification Kit MAK266 (Sigma-Aldrich) according to the instructions of the manufacturer.

### 4.6. Quantification of Plasma Free Fatty Acid Levels

Plasma free fatty (FFA) levels were determined using the Free Fatty Acids Quantification Kit (Sigma-Aldrich, St. Louis, MO, USA) according to the instructions of the manufacturer [53].

### 4.7. Determination of Plasma Levels of Insulin, Adiponectin, and Tumor Necrosis Factor-α

Plasma insulin levels in C57BL/6J LDLr^−/−^ mice were measured using the Ultra Sensitive Mouse Insulin enzyme-linked immunosorbent assay (ELISA) Kit (Crystal Chem, Elk Grove Village, IL, USA). Murine plasma adiponectin was determined by ELISA according to the instructions of the manufacturer (Thermo Fisher Scientific, Vienna, Austria). Plasma tumor necrosis factor (TNF)-α levels were quantified using the Mouse TNF-α Platinum ELISA (Thermo Fisher Scientific, Vienna, Austria).

### 4.8. Histological Analyses of the Myocardium

Histological analyses were performed as described before [54,55,56]. After hemodynamic analyses, C57BL/6J LDLr^−/−^ mice were perfused via the abdominal aorta with phosphate-buffered saline and hearts were arrested in diastole by KCl (100 μL; 0.1 mol/L), followed by perfusion fixation with 1% paraformaldehyde in phosphate-buffered saline. Thereafter, hearts were post-fixated overnight in 1% paraformaldehyde and embedded in paraffin. Cross-sections of 6 μm thickness at 130 μm spaced intervals were made, extending from the apex to the basal part of the left ventricle. Comparative sections were analyzed for all histological analyses by using the same slide numbers (1–40 from apex to base) and cross-section numbers (1–10).

To measure collagen content in the interstitium, Sirius Red staining was performed as described by Junqueira et al. [57]. Sirius Red polarization microscopy on a Leica RBE microscope with KS300 software (Zeiss) was applied to quantify thick, tightly-packed mature collagen fibers as orange-red birefringent and loosely-packed, less cross-linked, and immature collagen fibers as yellow-green birefringent. The collagen-positive area was normalized to the left ventricular wall area and was expressed as percentage. Any perivascular fibrosis was excluded from this analysis. Perivascular fibrosis was quantified as the ratio of the fibrosis area surrounding the vessel to the total vessel area. Two mid-ventricular sections were studied per animal [54].

Cardiomyocyte hypertrophy was analyzed on paraffin sections stained with rabbit anti-mouse laminin (Sigma; 1/50) by measuring the cardiomyocyte cross-sectional area (μm^2^) of at least 200 randomly selected cardiomyocytes in the left ventricular myocardium. Capillary density in the myocardium was determined on CD31-stained sections using rat anti-mouse CD31 antibodies (BD; 1/500). Relative vascularity was calculated as the ratio of capillary density to cardiomyocyte density divided by the cardiomyocyte cross-sectional area [58] and is expressed in µm^−2^. Two mid-ventricular cross-sections were analyzed per mouse [59,60].

Immunostaining for 3-nitrotyrosine was performed with rabbit anti-nitrotyrosine antibodies (Merck Millipore, Overijse, Belgium; dilution 1/250).

### 4.9. Histological Analysis of Gonadal Fat Pad

Hematoxylin and eosin staining was performed on paraffin sections with a thickness of 10 μm. The gonadal adipocyte cross-sectional area (µm^2^) and density (number/mm^2^) were determined on images taken at 100x magnification. The computerized image analysis was performed using KS300 software (Zeiss).

### 4.10. Quantification of Liver Steatosis

Liver paraffin sections (with a thickness of 10 µm were stained for hematoxylin and eosin. Images were taken at ×200 magnification and the area of steatosis was indirectly quantified as unstained/white area. Steatosis area was normalized to the total selected area and was expressed as percentage.

### 4.11. Exercise Treadmill Testing

A motor-driven treadmill (Treadmill Simplex II, Columbus Instruments, Columbus, OH, USA) was applied to evaluate the exercise capacity in mice [52,61]. C57BL/6J LDLr^−/−^ mice were familiarized with running on a motorized treadmill for one week. To quantify endurance capacity, mice started running on a 10° incline at an initial speed of 10 m/min, which was increased by 1 m/min every minute until the mouse resides on the stimulus plate (pulse grill) for ≥5 s. At this point, the mouse was immediately removed from the treadmill. The total exercise time was recorded as the elapsed time to exhaustion (min) and was then converted to distance (m), which was the end-point. Mice were subjected to tail snip before and after exercise the tolerance test for a lactate analysis (EKF Diagnostics, Penarth, Cardiff, UK).

### 4.12. Statistical Analysis

No mice died during the study. At the end of the study, data of all surviving mice were included in the analysis. Investigators who performed endpoint analyses were blinded to group allocation. Unblinding of animal numbers corresponding to specific allocation groups was performed at the completion of measurements.

Statistical analysis was performed as outlined before [55,56]. Data are expressed as means ± standard error of the means (SEM). Minimally required sample size calculation (*n* = 13) for proving the effect of AAV8-LDLr gene transfer on hemodynamic parameters in HSHF diet mice was based on a statistical power of 90%, a two-sided cut-off value of statistical significance of 0.05, a difference of main hemodynamic parameters at the population level of 20%, and a standard deviation at population level at 16% of the average of population means. Parameters between four groups were compared by one-way analysis of variance followed by Bonferroni multiple comparisons post-test for comparing SC diet groups, HSHF diet groups, and SC diet groups versus the respective HSHF diet groups using GraphPad InStat (GraphPad Software, San Diego, CA, USA). When the assumption of sampling from populations with identical standard deviations was not met, a logarithmic transformation was performed. When the assumption of sampling from populations with Gaussian distributions was not met, a Kruskal-Wallis test was performed, followed by a Dunn’s multiple comparisons post-test. A two-sided *p*-value of less than 0.05 was considered statistically significant.

## Figures and Tables

**Figure 1 ijms-20-02222-f001:**
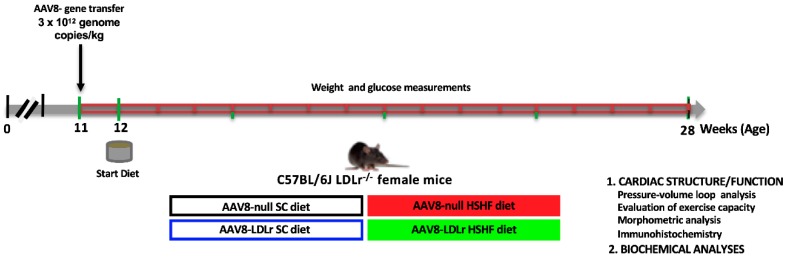
Schematic representation of the study design.

**Figure 2 ijms-20-02222-f002:**
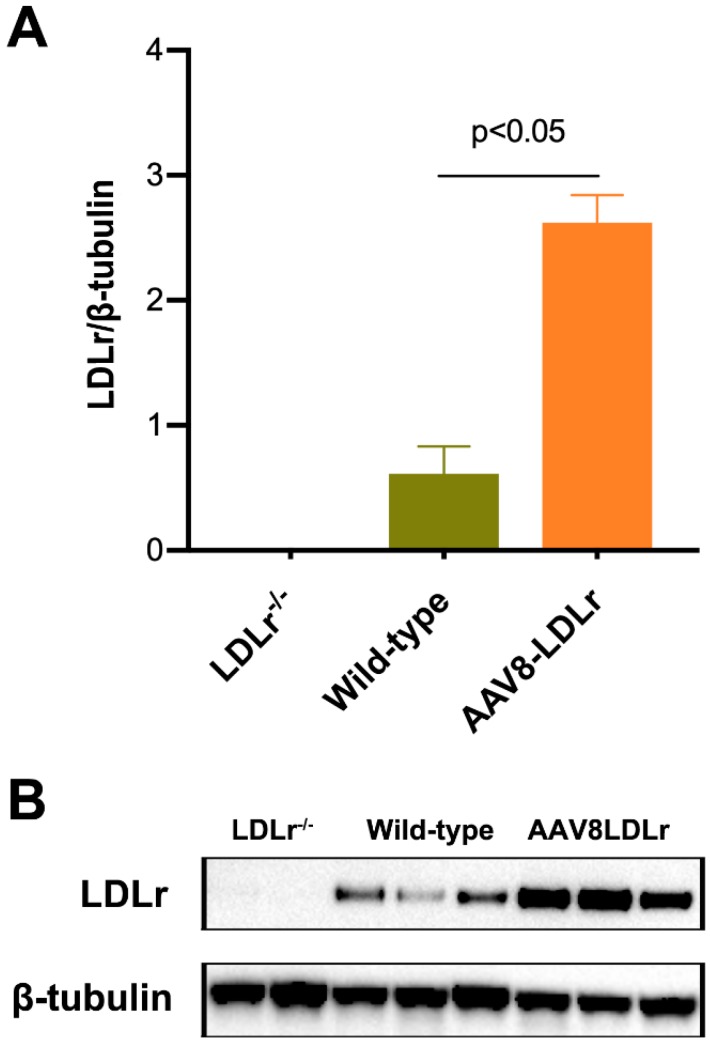
Quantification of murine LDLr expression in the liver. Bar graph (**A**) illustrating murine LDLr protein levels quantified by western blot in the liver of C57BL/6J LDLr^−/−^ mice (*n* = 2), of wild-type C57BL/6J mice (*n* = 3), and of AAV8-LDLr C57BL/6 LDLr^−/−^ mice (*n* = 3) 17 weeks after gene transfer with 3 x 10^12^ genome copies/kg of AAV8-LDLr. All protein levels were normalized to the ß-tubulin protein level. Image of western blot is shown in panel (**B**). The first two lanes correspond to C57BL/6J LDLr^−/−^ mice, the next three lanes illustrate wild type C57BL/6 J mice, and the final three lanes correspond to AAV8-LDLr-treated C57BL/6 LDLr^−/−^ mice.

**Figure 3 ijms-20-02222-f003:**
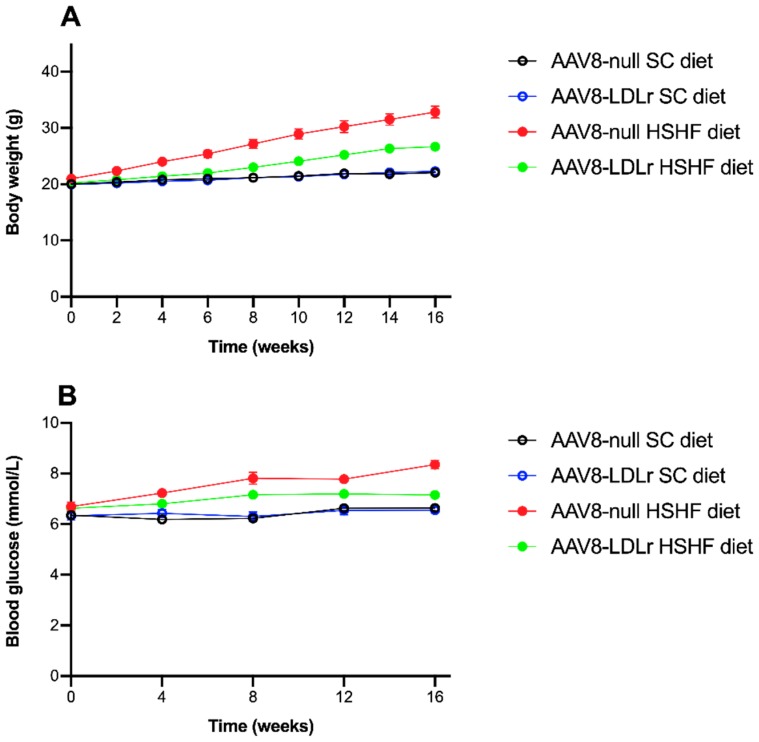
Time course of body weight (**A**) and blood glucose levels (**B**) in C57BL/6J LDLr^−/−^ mice fed a standard chow (SC) diet or a high-sucrose/high-fat (HSHF) diet. Week 0 in panels (A) and (B) corresponds to the age of 12 weeks, the start of the HSHF diet. All data represent mean ± SEM (*n* = 24 for SC diet groups; *n* = 36 for HSHF diet groups).

**Figure 4 ijms-20-02222-f004:**
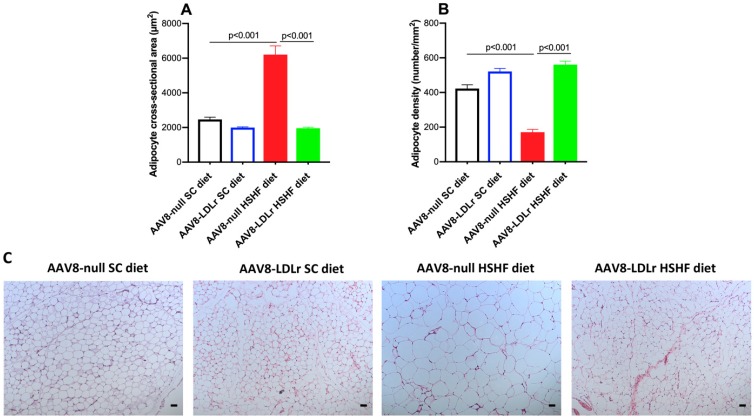
AAV8-LDLr gene transfer abrogates adipocyte hypertrophy induced by the HSHF diet. Adipocyte cross-sectional area (**A**) and adipocyte density (**B**) in female C57BL/6J LDLr^−/−^ mice at 16 weeks after the start of the diet. Data are expressed as means ± SEM (*n* = 5). (**C**) This panel contains representative photomicrographs illustrating haematoxylin and eosin-stained adipocytes of the gonadal fat pad. The scale bar represents 50 µm.

**Figure 5 ijms-20-02222-f005:**
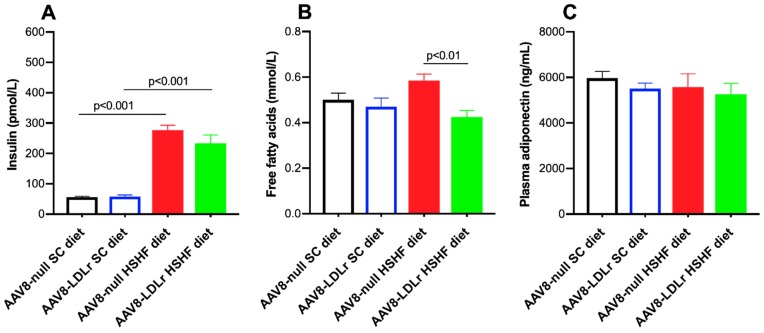
Plasma insulin (**A**), free fatty acids (**B**), and adiponectin (**C**) levels at 16 weeks after the start of the diet. All data represent means ± SEM (*n* = 10).

**Figure 6 ijms-20-02222-f006:**
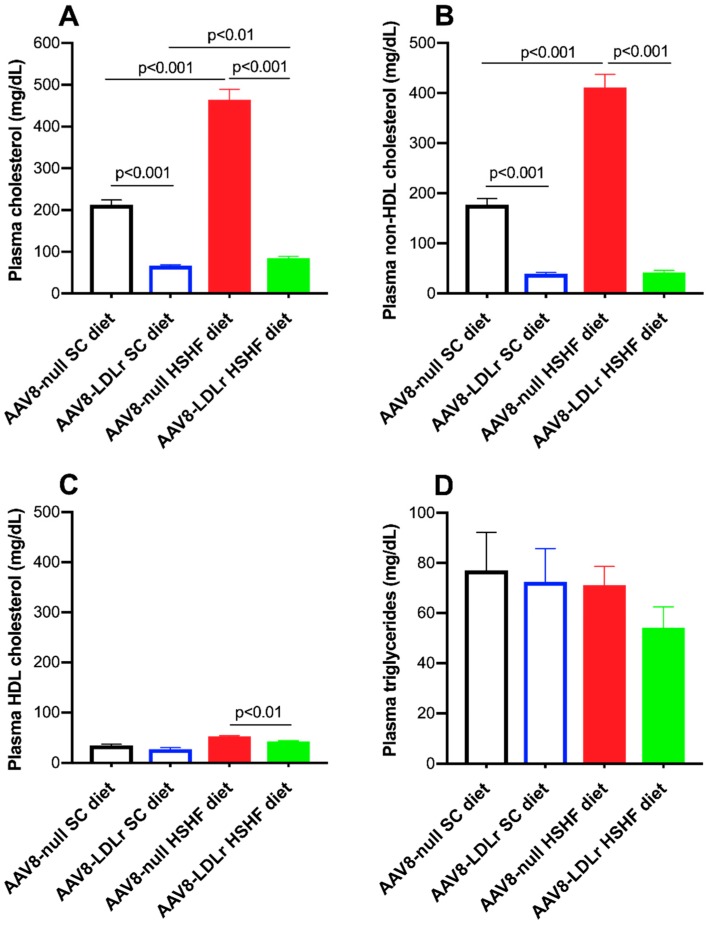
Total cholesterol (**A**), non-HDL cholesterol (**B**), HDL cholesterol plasma levels (**C**), and plasma triglyceride levels (**D**) in female C57BL/6J LDLr^−/−^ mice at 16 weeks after the start of the diet. All data are represented as means ± SEM (*n* = 10).

**Figure 7 ijms-20-02222-f007:**
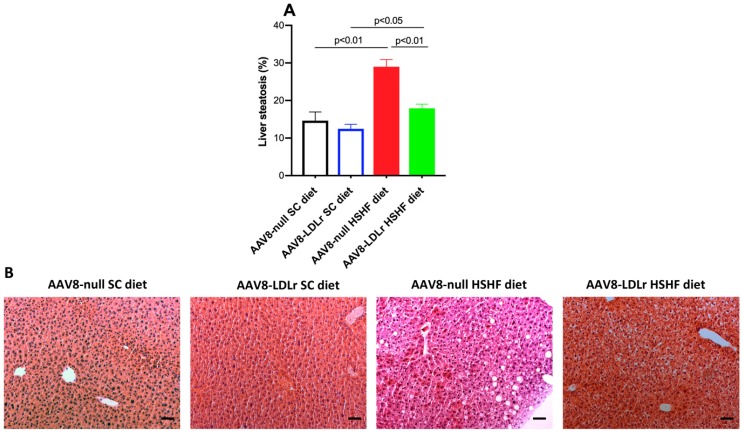
AAV8-LDLr gene transfer attenuates liver steatosis induced by the HSHF diet. Liver steatosis expressed as percentage of surface area in female C57BL/6J LDLr^−/−^ mice at 16 weeks after the start of the diet is shown in panel (**A**). Data are expressed as means ± SEM (*n* = 5). Panel (**B**) contains representative photomicrographs illustrating hematoxylin- and eosin-stained liver tissue. The scale bar represents 50 µm.

**Figure 8 ijms-20-02222-f008:**
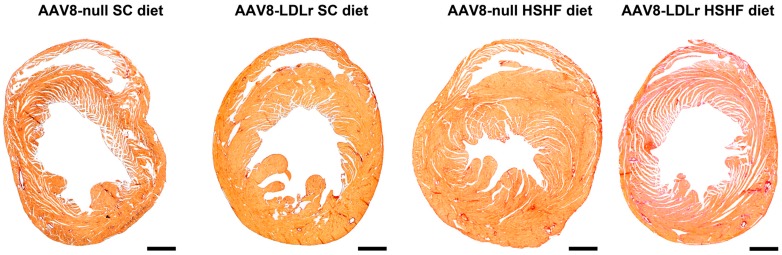
Representative Sirius Red-stained heart cross-sections of SC diet and HSHF diet mice. The scale bar represents 1 mm.

**Figure 9 ijms-20-02222-f009:**
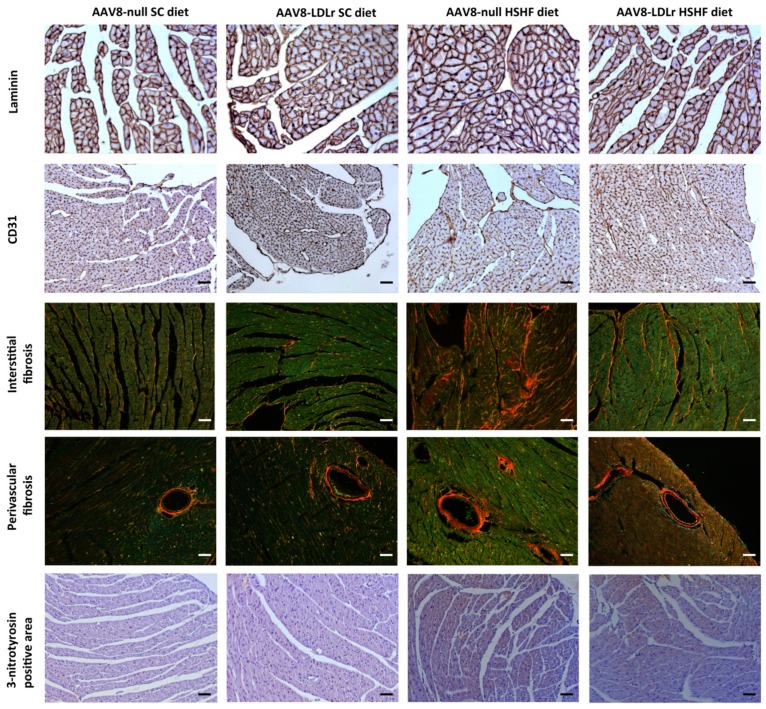
Immunohistochemical and histochemical analysis of the myocardium of SC diet and HSHF diet mice. Representative photomicrographs show laminin-stained cardiomyocytes, CD31-positive capillaries, Sirius Red-stained collagen, and 3-nitrotyrosin. The scale bar represents 50 µm.

**Figure 10 ijms-20-02222-f010:**
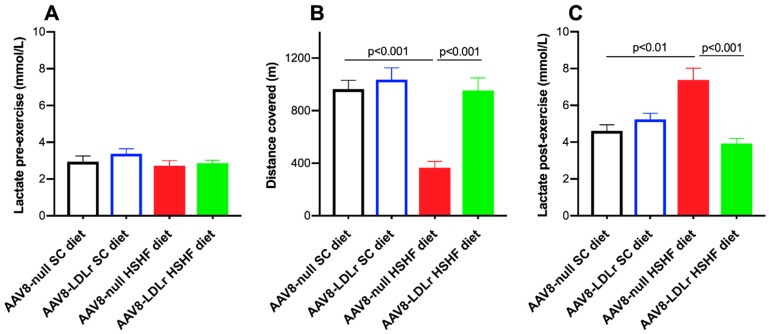
Lactate pre-exercise (**A**), distance covered (**B**), and lactate post-exercise (**C**) in C57BL/6 LDLr^−/−^ mice fed the SC diet or the HSHF diet. All data represent means ± SEM (*n* = 11 for SC diet mice, *n* = 15 for HSHF diet mice).

**Figure 11 ijms-20-02222-f011:**
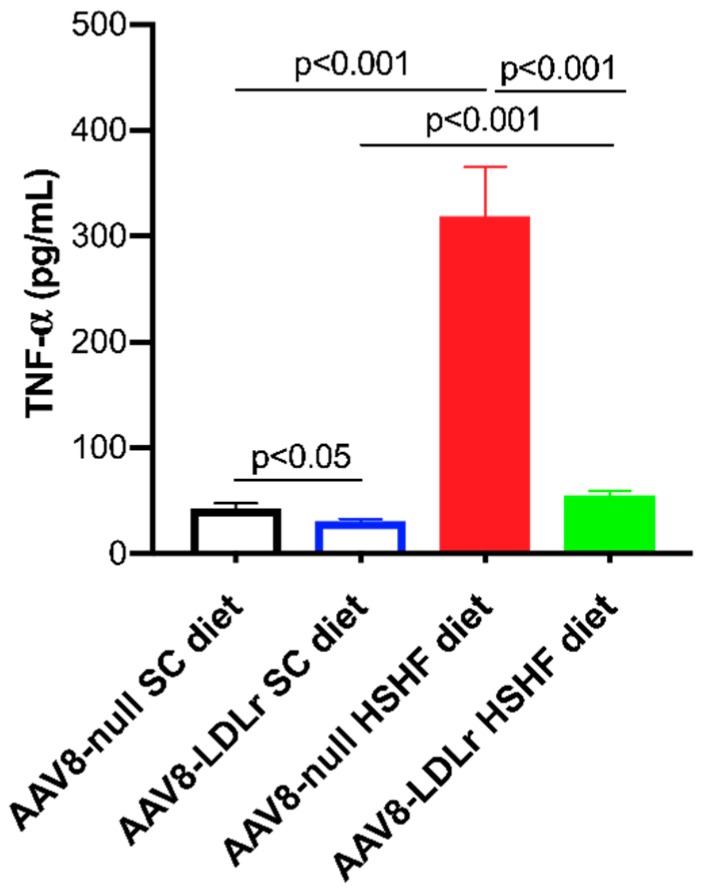
Plasma TNF-α concentration in C57BL/6J LDLr^−/−^ mice fed the SC diet or the HSHF diet and in C57BL/6J mice fed the SC diet or the HSHF diet. All data represent means ± SEM (*n* = 10).

**Table 1 ijms-20-02222-t001:** Organ and tissue weights in C57BL/6J LDLr^−/−^ mice fed the SC diet or the HSHF diet.

	AAV8-null SC Diet (*n* = 14)	AAV8-LDLr SC Diet (*n* = 12)	AAV8-null HSHF Diet (*n* = 15)	AAV8-LDLr HSHF Diet (*n* = 23)
Heart weight (mg)	118 ± 3	114 ± 2	140 ± 4 ^§§§^	118 ± 2 ***
Tibia length (mm)	17.4 ± 0.1	17.3 ± 0.1	17.5 ± 0.1	17.4 ± 0.1
Heart weight/tibia length (mg/mm)	6.82 ± 0.15	6.58 ± 0.08	8.01 ± 0.21 ^§§§^	6.79 ± 0.10 ***
Left ventricular weight (mg)	78.8 ± 1.8	75.5 ± 1.0	97.2 ± 2.5 ^§§§^	81.0 ± 1.1 ^§§^***
Right ventricular weight (mg)	22.0 ± 0.9	24.3 ± 1.0	29.6 ± 2.4 ^§§^	22.5 ± 0.7 **
Lung weight (mg)	140 ± 6	140 ± 2	166 ± 4 ^§§§^	145 ± 3 ***
Liver weight (mg)	917 ± 21	896 ± 22	1230 ± 50 ^§§§^	1060 ± 40 ^§^**
Kidney weight(mg)	245 ± 5	257 ± 8	300 ± 8 ^§§§^	259 ± 3 ***
Spleen weight (mg)	69.2 ± 1.6	75.9 ± 1.9	95.0 ± 2.9 ^§§§^	77.8 ± 1.9 ***

Mice were sacrificed and organs and tissues were collected at the age of 28 weeks, 16 weeks after the start of the HSHF diet. Kidney weight represents the weight of both kidneys together. All data are expressed as means ± SEM. ^§^: *p* < 0.05; ^§§^: *p* < 0.01; ^§§§^: *p* < 0.001 versus the respective SC diet group. **: *p* < 0.01; ***: *p* < 0.001 versus HSHF diet AAV8-null.

**Table 2 ijms-20-02222-t002:** Morphometric and histological parameters of the left ventricular myocardium in C57BL/6J LDLr^−/−^ mice fed the SC diet or the HSHF diet.

	AAV8-null SC Diet (*n* = 11)	AAV8-LDLr SC Diet (*n* = 11)	AAV8-null HSHF Diet (*n* = 20)	AAV8-LDLr HSHF Diet (*n* = 21)
Left ventricular wall area (mm^2^)	9.28 ± 0.28	8.41 ± 0.33	11.5 ± 0.3 ^§§§^	9.92 ± 0.33 ^§§^**
Anterior wall thickness (µm)	1060 ± 40	980 ± 33	1340 ± 40 ^§§§^	1130 ± 30 ^§§^***
Cardiomyocyte cross-sectional area (µm^2^)	189 ± 7	197 ± 8	326 ±17 ^§§§^	219 ± 9 ***
Cardiomyocyte density (number/mm^2^)	3770 ± 120	3540 ± 110	2460 ± 120 ^§§§^	3490 ± 150 ***
Capillary density (number/mm^2^)	6040 ± 300	6580 ± 130	4770 ± 250 ^§§^	5790 ± 200 ^§^**
Relative vascularity (µm^−2^)	0.00864 ± 0.00058	0.00965 ± 0.00032	0.00633 ± 0.00041 ^§§^	0.00784 ± 0.00029 ^§§^**
Interstitial fibrosis (%)	1.86 ± 0.18	2.01 ± 0.20	3.81 ± 0.31 ^§§§^	2.38 ± 0.17 ***
Perivascular fibrosis (ratio)	0.456 ± 0.048	0.436 ± 0.048	0.522 ± 0.023	0.408 ± 0.044 *
3-nitrotyrosine positive area (%)	1.51 ± 0.15	1.90 ± 0.16	5.11 ± 0.45 ^§§§^	3.55 ± 0.36 ^§§^*

Histological and morphometric analyses were performed at the age of 28 weeks, 16 weeks after the start of the HSHF diet. All data are expressed as means ± SEM. ^§^: *p* < 0.05; ^§§^: *p* < 0.01; ^§§§^: *p* < 0.001 versus respective SC diet group. *: *p* < 0.05; **: *p* < 0.01; ***: *p* < 0.001 versus HSHF diet AAV8-null.

**Table 3 ijms-20-02222-t003:** Overview of hemodynamic data in C57BL/6 LDLr^−/−^ mice fed the SC diet or the HSHF diet.

	AAV8-null SC Diet (*n* = 10)	AAV8-LDLr SC Diet (*n* = 7)	AAV8-null HSHF Diet (*n* = 14)	AAV8-LDLr HSHF Diet (*n* = 13)
Heart rate (bpm)	607 ± 12	580 ± 15	600 ± 16	580 ± 17
P_max_ (mm Hg)	99.9 ± 1.9	102 ± 1	87.7 ± 2.4 ^§§^	97.5 ± 3.0 *
P_es_ (mm Hg)	95.2 ± 2.4	94.3 ± 1.6	80.2 ± 2.5 ^§§^	90.5 ± 3.4 *
dP/dt_max_ (mmHg/ms)	11.9 ± 0.3	14.2 ± 0.4 ^°°^	8.46 ± 0.35 ^§§§^	11.4 ± 1.2 *
PRSW (mm Hg)	74.0 ±1.6	77.8 ± 5.1	59.4 ± 4.1 ^§^	84.5 ± 5.6 **
E_es_ (mmHg/µl)	6.67 ± 0.63	8.16 ± 0.82	4.69 ± 0.55 ^§^	6.40 ± 0.47 *
P_min_ (mm Hg)	1.82 ± 0.44	0.415 ± 0.494	1.99 ± 0.57	1.47 ± 0.43
P_ed_ (mm Hg)	5.02 ± 0.37	4.23 ± 0.44	5.24 ± 0.53	5.36 ± 0.70
dP/dt_min_ (mmHg/ms)	−9.93 ± 0.55	−10.4 ± 0.5	−7.93 ± 0.36 ^§§^	−9.65 ± 0.51 *
Tau (ms)	5.64 ± 0.16	5.45 ± 0.21	7.14 ± 0.22 ^§§§^	5.74 ± 0.22 ***
Slope EDPVR (mmHg/µl)	0.523 ± 0.070	0.487 ± 0.155	0.932 ± 0.103 ^§§^	0.276 ± 0.081 ***
EDV (µl)	31.5 ± 1.5	27.6 ± 2.0	25.1 ± 1.4 ^§^	29.2 ± 2.1
ESV (µl)	13.6 ± 1.0	9.15 ± 1.00 ^°^	12.2 ± 1.2	12.7 ± 1.2
Stroke volume (µl)	17.9 ± 0.9	18.4 ± 1.3	12.9 ± 0.7 ^§§§^	16.5 ± 1.2 *
Ejection fraction (%)	57.2 ± 2.1	67.0 ± 2.0 ^°°^	52.1 ± 2.5	56.9 ± 1.8 ^§§^
Cardiac output (ml/min)	10.9 ± 0.6	10.7 ± 0.8	7.76 ± 0.46 ^§§§^	9.69 ± 0.72 *
Stroke work (mmHg.µl)	1420 ± 70	1540 ± 110	898 ± 46 ^§§§^	1310 ± 120 **
dV/dt_max_ (µl/s)	712 ± 22	839 ± 86	509 ± 52 ^§§^	554 ± 51 ^§^
dV/dt_min_ (µl/s)	−790 ± 44	−992 ± 70 ^°^	−587 ± 46 ^§§^	−672 ± 40 ^§§^
E_a_ (mmHg/µl)	5.42 ± 0.38	5.19 ± 0.40	6.55 ± 0.52	5.75 ± 0.36
E_a_/E_es_	0.865 ± 0.086	0.687 ± 0.101	1.79 ± 0.35 ^§^	0.966 ± 0.081 *

Hemodynamic measurements were performed at the age of 28 weeks, 16 weeks after the start of the HSHF diet. P_max_: maximum systolic pressure. P_es_: end-systolic pressure. dP/dt_max_: peak rate of isovolumetric contraction. PRSW: preload recruitable stroke work. E_es_: end-systolic elastance. P_min_: minimum diastolic pressure. P_ed_: end-diastolic pressure. dP/dt_min_: peak rate of isovolumetric relaxation. Tau: time constant of isovolumetric relaxation. EDPVR: end diastolic pressure–volume relationship. EDV: end-diastolic volume. ESV: end-systolic volume. dV/dt_max_: peak filling rate. dV/dt_min_: peak emptying rate. E_a_: arterial elastance. E_a_/E_es_: ventriculo–arterial coupling ratio. All data are expressed as means ± SEM. ^°^: *p* < 0.05; ^°°^: *p* < 0.01 versus SC diet AAV8-null. ^§^: *p* < 0.05; ^§§^: *p* < 0.01; ^§§§^: *p* < 0.001 versus respective SC diet group. *: *p* < 0.05; **: *p* < 0.01; ***: *p* < 0.001 versus HSHF diet AAV8-null.

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
