# Peer review of "Cholesterol-Lowering Gene Therapy Prevents Heart Failure with Preserved Ejection Fraction in Obese Type 2 Diabetic Mice"

_ijms, 2019, doi:10.3390/ijms20092222_

Reviewer 1 Report

The authors explore a cholesterol-based therapy in a experimental mouse model of heart failure. Authors have done extensive work on characterizing the impact of LDLR overexpression on the cardiac phenotype of  LDLR-deficient mice. However, the link between the favorable impact by LDLR overexpression in the liver of AAV8-LDLR HSHF mice on body weight prevention and its relationship with improvement in heart remodeling and normalized cardiac function is poorly explored and discussed.

The ms is well written and it looks very interesting, but I would recommend to accept it after major revision.

Main points:

-- > Why the weight gain of LDLR-deficient mice was decreased? It looks significantly reduced at the end of the treatment in AAV8-LDLR HSHF mice vs. AAV8-null HSHF mice (Figure 2). In the discussion section (lines 283-288) it is also observed in C57BL/6J mice, which led the authors to hypothesized that this phenotype to be caused by the absence of the LDLR “… in non-hepatocytes?? has a significant impact on global metabolism”, however, overexpression of LDLR is liver-specific. Its favorable impact on the body weight gain is poorly described in the present version of the ms. and its relationship with improved heart remodeling and function is poorly discussed. In this regard, how the authors may explain this change in the body weight gain? Which are the mechanisms potentially involved? Were differences in food intake observed? Were adiposity (fad pads) weighed at necropsy? These (gross) parameters should appear in a table along with the body weight or even mentioned in the body text in a final version of the ms.

-- > Circulating free fatty acid were significantly decreased in AAV8-LDLR HSHF mice vs. AAV8-null HSHF mice (Figure 3): it might reflect improved peripheral insulin resistance and might be concomitant to favorable weight gain prevention observed in AAV8-LDLR HSHF mice. Did the authors carry out a glucose tolerance test to directly check it? Were the plasma levels of free fatty acids in AAV8-null HSHF mice significantly different from AAV8-null or –LDLR SC mice?

--> Liver weight and other organs of AAV8-LDLR HSHF mice were reduced compared with AAV8-null HSHF mice. Was it found associated with a favorable reduction in lipid accumulation of these tissues?

--> Why the authors did not show the gene expression levels of Ldlr or LDLr protein abundance in target tissues (liver) of LDLR-deficient mice? Their levels should be included in revised form of the ms.

Author Response

The authors thank the first reviewer for the constructive analysis of our manuscript.

The authors explore a cholesterol-based therapy in an experimental mouse model of heart failure. Authors have done extensive work on characterizing the impact of LDLR overexpression on the cardiac phenotype of LDLR-deficient mice. However, the link between the favorable impact by LDLR overexpression in the liver of AAV8-LDLR HSHF mice on body weight prevention and its relationship with improvement in heart remodeling and normalized cardiac function is poorly explored and discussed.

The ms is well written and it looks very interesting, but I would recommend to accept it after major revision.

Main points:

-- > Why the weight gain of LDLR-deficient mice was decreased? It looks significantly reduced at the end of the treatment in AAV8-LDLR HSHF mice vs. AAV8-null HSHF mice (Figure 2). In the discussion section (lines 283-288) it is also observed in C57BL/6J mice, which led the authors to hypothesized that this phenotype to be caused by the absence of the LDLR “… in non-hepatocytes?? has a significant impact on global metabolism”, however, overexpression of LDLR is liver-specific. Its favorable impact on the body weight gain is poorly described in the present version of the ms. and its relationship with improved heart remodeling and function is poorly discussed. In this regard, how the authors may explain this change in the body weight gain? Which are the mechanisms potentially involved? Were differences in food intake observed? Were adiposity (fad pads) weighed at necropsy? These (gross) parameters should appear in a table along with the body weight or even mentioned in the body text in a final version of the ms.

This study was designed to develop a model of heart failure with preserved ejection fraction (HFpEF) and not as a metabolic study. The difference in weight between AAV8-LDLr HSHF diet mice and AAV8-null HSHD diet mice was a startling observation. 

New data that have been added in the Results section are relevant for the subsequent discussion of the body weight difference. These data relate to murine LDLr expression and to adipocyte hypertrophy.

The following results can be found on page lines 73-77 of the revised manuscript:

‘Murine LDLr expression in the liver C57BL/6 LDLr-/- mice, wild-type C57BL/6J mice, and C57BL/6J LDLr-/- mice treated with 3 x 1012 genome copies/kg of AAV8-LDLr is illustrated in Figure 2. Murine LDLr was undetectable in C57BL/6J LDLr-/- mice. Murine LDLr expression 17 weeks after gene transfer with AAV8-LDLr in C57BL/6J LDLr-/- mice was 4.26-fold (p<0.05) higher compared to wild-type C57BL/6J mice.’

The methodology of the western blot has been added in the revised manuscript on page lines 423-431:

‘4.3. Quantification of murine LDLr expression in the liver by western blot

Liver tissue samples were isolated and immediately frozen in liquid nitrogen and stored at -80°C. The extraction, blotting, and protein quantification procedures have been described in detail before[1]. The primary antibody for detection of the murine LDLr was obtained from Abcam (Cambrdige, UK). The primary antibody for the quantification of b-tubulin was purchased from Cell Signalling Technologies (Beverly, MA, USA), Protein expression was detected with Super signal west pico chemilumninescent reagents (Thermo Scientific, Rockford, IL, USA) and quantified using Image lab TM Analyzer software (Bio-Rad laboratories N.V.).  All protein levels were normalized to the b-tubulin protein level’.

Data on adipocyte hypertrophy can be retrieved on page lines 105-113 of the revised manuscript:

‘To evaluate the effect of AAV8-LDLr gene transfer on adiposity induced by the HSHF diet, histological analysis of the gonadal fat pad was performed (Figure 4). The adipocyte cross-sectional area was 2.52-fold (p<0.001) higher in AAV8-null HSHF diet mice than in AAV8-null SC diet mice. AAV8-LDLr gene transfer abrogated adipocyte hypertrophy induced by the HSHF diet (Figure 4A). The adipocyte cross-sectional area was reduced by 68.4% (p<0.001) in AAV8-LDLr HSHF diet mice compared to AAV8-null HSHF diet mice. Adipocyte density was reduced by 59.5% (p<0.001) in AAV8-null HSHF diet mice compared to AAV8-null SC diet mice and was not reduced at all in AAV8-LDLr HSHF diet mice.’

The methodology concerning histological analysis of the gonadal fat pad can be retrieved on page lines 489-493:

‘4.9. Histological analysis of gonadal fat pad

Haematoxylin and eosin staining was performed on paraffin sections with a thickness of 10 μm. Gonadal adipocyte cross-sectional area (µm2) and density (number/mm2) were determined on images taken at 100x magnification. Computerized image analysis was performed using KS300 software (Zeiss).’

We noticed that we had omitted the daily food intake data that we had collected. The following sentence has been incorporated in the Results section on page lines 93-95:

‘Daily food intake measured during the entire duration of the experiment was not significantly different between AAV8-null HSHF diet mice (5.93 ± 0.37 g/mouse/day) and AAV8-LDLr HSHF diet mice (6.23 ± 0.33 g/mouse/day)’.

We have rewritten and expanded this part of the Discussion (lines 312-340) as follows:

‘A surprising finding in this study is that AAV8-LDLr gene therapy attenuated the weight gain in mice receiving the HSHF diet. This difference is not explained by a difference in food intake since daily food intake was not different between AAV8-null HSHF diet mice and AAV8-LDLr HSHF diet mice. In general, lipoprotein metabolic pathways are a pivotal contributor to the development of obesity. Modulation of lipolysis, receptor-mediated clearance of triglyceride-rich lipoproteins, and cross-talk between the liver and adipose tissue are major regulators of energy expenditure, whole-body homeostasis, and body weight. Apolipoprotein E is a key protein in the clearance of chylomicron remnants, LDL, and very low-density lipoproteins (VLDL). Apolipoprotein E deficient mice are resistant to diet-induced obesity, insulin resistance, and glucose intolerance[2-4]. Adipocyte-specific inactivation of the multifunctional receptor LDL receptor-related protein-1 (LRP1) resulted in delayed postprandial lipid clearance, reduced body weight, smaller fat stores, resistance to dietary fat-induced obesity, improved glucose tolerance, and elevated energy expenditure due to enhanced muscle thermogenesis[5]. The LDLr is also expressed in adipocytes[6]. Uptake and degradation of VLDL, intermediate density lipoprotein (IDL), and LDL particles may subsequently lead to the release of molecules with signalling properties[7,8]. Binding of apoB100-containing LDL to adipocytes via the LDL receptor inhibits intracellular noradrenaline-induced lipolysis in adipocytes[9].

Modulation of body weight by the LDLr may be tissue-dependent. On the one hand, absence of the LDLr in adipocytes of C57BL/6J LDLr-/- mice may attenuate the increase in adiposity when placed on an HSHF diet. Indeed, body weight in AAV8-null C57BL/6J LDLr-/- HSHF diet mice was 9.38% (p<0.05) lower than in wild-type C57BL/6J HSHF diet mice with exactly the same genetic background (unpublished data). This difference is also consistent with observations in humans showing that body mass index in patients with heterozygous familial hypercholesterolemia is significantly lower compared to unaffected relatives[10]. On the other hand, the supraphysiological expression level of the murine LDLr in parenchymal liver cells of AAV8-LDL C57BL/6J LDLr-/- HSHF diet mice may further attenuate weight gain by potentiation of postprandial catabolism of triglyceride-rich lipoproteins and may explain the significantly lower body weight and the absence of adipocyte hypertrophy in AAV8-LDL C57BL/6J LDLr-/- HSHF diet mice compared to AAV8-null C57BL/6J LDLr-/- HSHF diet mice.’

We did and do not claim that the difference in body weight must be on the causal pathway between treatment (AAV8-LDLr gene transfer) and cardiovascular phenotype. This is a topic of speculation. Rather, it is the vision of the authors that the effect of treatment on cardiovascular phenotype is independent of the weight difference. This claim cannot be proven nor can it be refuted since causal inference is restricted to treatment allocation (AAV8-LDLr or AAV8-null) that was performed randomly. What is on the causal pathway between gene transfer and cardiovascular phenotype cannot be established with certainty.

-- > Circulating free fatty acid were significantly decreased in AAV8-LDLR HSHF mice vs. AAV8-null HSHF mice (Figure 3): it might reflect improved peripheral insulin resistance and might be concomitant to favorable weight gain prevention observed in AAV8-LDLR HSHF mice. Did the authors carry out a glucose tolerance test to directly check it? Were the plasma levels of free fatty acids in AAV8-null HSHF mice significantly different from AAV8-null or –LDLR SC mice?

As stated before, the primary objective of the study was to evaluate the effect of AAV8-LDLr on the development of HFpEF and we did not perform a glucose tolerance test. The glucose and insulin data suggest that insulin sensitivity is higher in AAV8-LDLr HSHF diet mice than in in AAV8-null HSHF diet mice since glucose levels were lower in the former. It is reasonable to assume that the difference in plasma free fatty acids reflects improved insulin sensitivity but at the same time, free fatty acids themselves may have an impact on insulin sensitivity. The statistical difference as indicated in the Figure is correct. The difference between free fatty acids between AAV8-null SC diet mice and AAV8-null HSHF diet mice did not reach statistical significance but a type II statistical error should be considered. For the discussion on body weight, the authors refer to the answer supra, which discusses the differential effect of LDL expression in adipose tissue and in hepatocytes on body weight.

--> Liver weight and other organs of AAV8-LDLR HSHF mice were reduced compared with AAV8-null HSHF mice. Was it found associated with a favorable reduction in lipid accumulation of these tissues?

To prove that liver steatosis was reduced in AAV8-LDLr HSHF diet mice compared to AAV8-null HSHF diet mice, quantification was performed on paraffin sections. Data on liver steatosis can be retrieved in the Results section on page lines 163-167 of the revised manuscript:

‘The degree of liver steatosis was 1.98-fold (p<0.01) higher in AAV8-null HSHF diet mice than in AAV8-null SC diet mice (Figure 7A). Liver steatosis was reduced by 38.1% (p<0.01) in AAV8-LDLr-HSHF diet mice compared to AAV8-null HSHF diet mice. Representative photomicrographs illustrating haematoxylin and eosin-stained liver tissue are shown in Figure 7B’.

The following paragraph has been added in the Results section lines (495-499):

‘4.10. Quantification of liver steatosis

Liver paraffin sections (with a thickness of 10 µm were stained for haematoxylin and eosin. Images were taken at x200 magnification and the area of steatosis was indirectly quantified as unstained/white area. Steatosis area was normalized to the total selected area and was expressed as percentage’.

A detailed study of kidney pathology is outside the scope of this study and is part of an ongoing collaboration with kidney pathologists and nephrologists at KU Leuven.

--> Why the authors did not show the gene expression levels of Ldlr or LDLr protein abundance in target tissues (liver) of LDLR-deficient mice? Their levels should be included in revised form of the ms.

We refer here to the answer that we have provided supra. The data are represented in the new Figure 2. The Legend to this Figure is as follows:

Figure 2. Quantification of murine LDLr expression in the liver. Bar graph (A) illustrating murine LDLr protein levels quantified by western blot in the liver of C57BL/6J LDLr-/- mice (n=2), of wild-type C57BL/6J mice (n=3), and of AAV8-LDLr C57BL/6 LDLr-/- mice (n=3) 17 weeks after gene transfer with 3 x 1012 genome copies/kg of AAV8-LDLr. All protein levels were normalized to the ß-tubulin protein level. Image of western blot is shown in panel B. The first two lanes correspond C57BL/6J LDLr-/- mice, the next three lanes illustrate wild type C57BL/6 J mice, and the final three lanes correspond to AAV8-LDLr-treated C57BL/6 LDLr-/- mice.’

REFERENCES

1.         Muthuramu, I.; Amin, R.; Postnov, A.; Mishra, M.; Aboumsallem, J.P.; Dresselaers, T.; Himmelreich, U.; Van Veldhoven, P.P.; Gheysens, O.; Jacobs, F., et al. Cholesterol-Lowering Gene Therapy Counteracts the Development of Non-ischemic Cardiomyopathy in Mice. Mol Ther 2017, 25, 2513-2525, doi:10.1016/j.ymthe.2017.07.017.

2.         Kypreos, K.E.; Karagiannides, I.; Fotiadou, E.H.; Karavia, E.A.; Brinkmeier, M.S.; Giakoumi, S.M.; Tsompanidi, E.M. Mechanisms of obesity and related pathologies: role of apolipoprotein E in the development of obesity. FEBS J 2009, 276, 5720-5728, doi:10.1111/j.1742-4658.2009.07301.x.

3.         Karagiannides, I.; Abdou, R.; Tzortzopoulou, A.; Voshol, P.J.; Kypreos, K.E. Apolipoprotein E predisposes to obesity and related metabolic dysfunctions in mice. FEBS J 2008, 275, 4796-4809, doi:10.1111/j.1742-4658.2008.06619.x.

4.         Hofmann, S.M.; Perez-Tilve, D.; Greer, T.M.; Coburn, B.A.; Grant, E.; Basford, J.E.; Tschop, M.H.; Hui, D.Y. Defective lipid delivery modulates glucose tolerance and metabolic response to diet in apolipoprotein E-deficient mice. Diabetes 2008, 57, 5-12, doi:10.2337/db07-0403.

5.         Hofmann, S.M.; Zhou, L.; Perez-Tilve, D.; Greer, T.; Grant, E.; Wancata, L.; Thomas, A.; Pfluger, P.T.; Basford, J.E.; Gilham, D., et al. Adipocyte LDL receptor-related protein-1 expression modulates postprandial lipid transport and glucose homeostasis in mice. J Clin Invest 2007, 117, 3271-3282, doi:10.1172/JCI31929.

6.         Kraemer, F.B.; Laane, C.; Park, B.; Sztalryd, C. Low-density lipoprotein receptors in rat adipocytes: regulation with fasting. Am J Physiol 1994, 266, E26-32, doi:10.1152/ajpendo.1994.266.1.E26.

7.         Herz, J.; Gotthardt, M.; Willnow, T.E. Cellular signalling by lipoprotein receptors. Curr Opin Lipidol 2000, 11, 161-166.

8.         Nykjaer, A.; Willnow, T.E. The low-density lipoprotein receptor gene family: a cellular Swiss army knife? Trends Cell Biol 2002, 12, 273-280.

9.         Skogsberg, J.; Dicker, A.; Ryden, M.; Astrom, G.; Nilsson, R.; Bhuiyan, H.; Vitols, S.; Mairal, A.; Langin, D.; Alberts, P., et al. ApoB100-LDL acts as a metabolic signal from liver to peripheral fat causing inhibition of lipolysis in adipocytes. PLoS One 2008, 3, e3771, doi:10.1371/journal.pone.0003771.

10.       Besseling, J.; Kastelein, J.J.; Defesche, J.C.; Hutten, B.A.; Hovingh, G.K. Association between familial hypercholesterolemia and prevalence of type 2 diabetes mellitus. Jama 2015, 313, 1029-1036, doi:10.1001/jama.2015.1206.

Reviewer 2 Report

To:

Editorial Board

Title: “Cholesterol-lowering Gene Therapy Prevents Heart Failure with Preserved Ejection Fraction in Obese Type 2 Diabetic Mice”

Dear Editor,

I read this manuscript and I think that:

-          The Introduction section of the paper is too long and offers some redundant paragraphs. The authors concentrated their attention on the influence of diabetes on heart. Little attention was on cholesterol lowering gene therapy. Please revise this section in order to better relate it to the aims of the work.

-          The authors should separately evaluate the influence of cholesterol and diabetes on the final outcomes. The combination of these two cardiovascular risk factors can be misleading.

-          The role of nutraceuticals on clinical outcomes should be considered. Diet can effectively impair the final outcomes. Please discuss the paper from Scicchitano P et al. Journal of Functional Foods 2014;6:11-32.

Author Response

The authors thank the second reviewer for the constructive analysis of our manuscript.

-          The Introduction section of the paper is too long and offers some redundant paragraphs. The authors concentrated their attention on the influence of diabetes on heart. Little attention was on cholesterol lowering gene therapy. Please revise this section in order to better relate it to the aims of the work.

In the revised version, the authors have omitted the first paragraph of the Introduction, which was indeed not strictly required to position the paper. The literature on cholesterol lowering gene therapy and heart failure is very limited since this field was only recently opened by Muthuramu et al.[1]. This study is mentioned in the second paragraph of the Discussion. The authors consider that this paragraph is an essential part of the Discussion. It does not make sense to discuss cholesterol lowering gene therapy in general (e.g. in relation to atherosclerotic cardiovascular diseases) in the Introduction.

-          The authors should separately evaluate the influence of cholesterol and diabetes on the final outcomes. The combination of these two cardiovascular risk factors can be misleading.

Causal inference is restricted to treatment allocation (AAV8-LDLr or AAV8-null) that was performed randomly. What is on the causal pathway between gene transfer and cardiovascular phenotype cannot be established with certainty. The authors point out that AAV8-LDLr gene transfer also improves cardiac function in standard chow (SC) diet mice but these mice do not display heart failure and have a lower degree of hypercholesterolemia than AAV8-null HSHF diet mice. Taken together, the global study design also provides information on beneficial effects of AAV8-LDLr gene transfer in the absence of type 2 diabetes mellitus.

The following sentence has been added in the Discussion on page lines 290-293:

‘Moreover, pressure-volume loop analysis indicated that several hemodynamic parameters were ameliorated in AAV8-LDLr SC diet mice compared to AAV8-null SC diet mice. Therefore, AAV8-LDLr gene transfer also improved cardiac function in the absence of type 2 diabetes mellitus.’

-          The role of nutraceuticals on clinical outcomes should be considered. Diet can effectively impair the final outcomes. Please discuss the paper from Scicchitano P et al. Journal of Functional Foods 2014;6:11-32.

The authors have carefully read and analyzed the content of this review. The topic of this review is the effect of nutraceuticals on dyslipidemia. First of all, the nutraceuticals do not selectively lower cholesterol. Secondly, cholesterol levels are a biomarker and do represent a clinically meaningful endpoint. Therefore, no evidence is provided that these compounds may be useful for treatment of HFpEF in the setting of hypercholesterolemia and diabetes. Taken together, the authors respectfully disagree with this suggestion of the reviewer. This review does not fit in the framework of the discussion of  this manuscript.

REFERENCES

1.         Muthuramu, I.; Amin, R.; Postnov, A.; Mishra, M.; Aboumsallem, J.P.; Dresselaers, T.; Himmelreich, U.; Van Veldhoven, P.P.; Gheysens, O.; Jacobs, F., et al. Cholesterol-Lowering Gene Therapy Counteracts the Development of Non-ischemic Cardiomyopathy in Mice. Mol Ther 2017, 25, 2513-2525, doi:10.1016/j.ymthe.2017.07.017.

Round  2

Reviewer 1 Report

The article has been improved but the favorable impact of overexpressing LDLr in the livers of LDLr-deficient mice still remains puzzling. Actually, hepatic overexpression of LDLr improved cardiomyopathy in HSHF diet mice, being this mainly attributed to its cholesterol-lowering effect. Although some of the findings are just startling observations, they may concevibly contribute to the favorable changes in cardiomyopathy signs shown by these mice. That said, I still have some concerns:

1) Insulin sensitivity in AAV8-LDLr HSHF diet mice was not improved, glucose levels were not significantly different from AAV8-null HSHF diet mice. Additionaly, weight gain of the mice fed the HSHF diet was not significantly higher than that shown by the SC diet mice. Only adipocyte hypertrophy is shown, but adiposity (?) is not shown (i.e., weights of fad pads). may it mean that it was not significantly increased. In contrast, the mice fed the HSHF diet developed hepatic steatosis. Importantly, hepatic steatosis was corrected by liver-specific overexpression of LDLr. In summary, the HSHF diet mice might be rather considered as a mouse model of hepatic steatosis? Therefore, the mention in the title of "..in Obese Type 2 Diabetic Mice" would not be appropriate. Also check the conclusion.

2) Consumption of HSHF diet made the mice to develop hepatic steatosis, mainly due to the increased dietary content in fructose. This may lead to elevations of LDL levels due to an inadequate clearance of these lipoproteins. The authors show that liver-specific overexpression of LDLr prevents HSHF diet-induced hepatic steatosis and leads to favorable evolution of cardiomyopathy in mice. Importantly, this does not occur in SC diet mice. This finding might suggest that the induction of hepatic steatosis as the key contributor of cardiomyopathy? Which might be the mechanism linking liver dysfunction and impaired myocardium? Is there any evidence showing such link in patients? If so, it was corrected upon cholesterol-lowering treatment (i.e., statins)?

Author Response

The authors thank the reviewer for the constructive analysis of our revised manuscript.

The article has been improved but the favorable impact of overexpressing LDLr in the livers of LDLr-deficient mice still remains puzzling. Actually, hepatic overexpression of LDLr improved cardiomyopathy in HSHF diet mice, being this mainly attributed to its cholesterol-lowering effect. Although some of the findings are just startling observations, they may concevibly contribute to the favorable changes in cardiomyopathy signs shown by these mice. That said, I still have some concerns:

1) Insulin sensitivity in AAV8-LDLr HSHF diet mice was not improved, glucose levels were not significantly different from AAV8-null HSHF diet mice. Additionaly, weight gain of the mice fed the HSHF diet was not significantly higher than that shown by the SC diet mice. Only adipocyte hypertrophy is shown, but adiposity (?) is not shown (i.e., weights of fad pads). may it mean that it was not significantly increased. In contrast, the mice fed the HSHF diet developed hepatic steatosis. Importantly, hepatic steatosis was corrected by liver-specific overexpression of LDLr. In summary, the HSHF diet mice might be rather considered as a mouse model of hepatic steatosis? Therefore, the mention in the title of "..in Obese Type 2 Diabetic Mice" would not be appropriate. Also check the conclusion.

In contrast to what the reviewer states, glucose levels are lower in AAV8-LDLr HSHF diet mice compared to AAV8-null HSHF diet mice. This is clear from Figure 3B and is explicitly stated in lines 97-99:

‘Blood glucose levels in AAV8-LDLr HSHF diet mice were significantly lower at 8 weeks (p<0.05), at 12 weeks (p<0.05), and at 16 weeks (p<0.0001) compared to AAV8-null HSHF diet mice.’

As we explained in the first rebuttal, the glucose and insulin data suggest that insulin sensitivity is higher in AAV8-LDLr HSHF diet mice than in in AAV8-null HSHF diet mice since glucose levels were lower in the former.

With regard to weight gain, we refer to Figure 3A and the corresponding text in lines 86-95. The HSHF diet induced an increase in body weight in both the AAV8-null HSHF diet mice and in the AAV8-LDLr HSHF diet mice compared to the respective SC diet groups. These body weight differences between HSHF diet groups and SC diet groups can only be explained by an increase in adipose tissue. After all, the difference in body weight is more than 10 g and more than 4 g for AAV8-null HSHF diet mice and AAV8-LDLr HSHF diet mice, respectively, compared to respective SC diet groups (Figure 3A). Tibia length in the four groups is the same and cumulative weight differences of the internal organs are not even close to 1 g.

In general, type 2 diabetes mellitus is a systemic disease and non-alcoholic fatty liver disease (NAFLD) is highly prevalent in subjects with obesity and type 2 diabetes mellitus[1]. The global prevalence of NAFLD in the world is as high as one billion[2]. In patients with type 2 diabetes mellitus without known history of cardiac disease, NAFLD was associated with early features of left ventricular diastolic dysfunction[3].

The following paragraph has been added in the Discussion lines 312-317:

‘Type 2 diabetes mellitus is a systemic disease that affects several organs. The HSHF diet model was characterized by the presence of hepatic steatosis, which is concordant with earlier observations[4].  Non-alcoholic fatty liver disease (NAFLD) is also highly prevalent in subjects with obesity and type 2 diabetes mellitus[1]. The global prevalence of NAFLD in the world is as high as one billion[2]. In patients with type 2 diabetes mellitus without known history of cardiac disease, NAFLD was associated with early features of left ventricular diastolic dysfunction[3].’

As we have stated before, causal inference is restricted to treatment allocation (AAV8-LDLr or AAV8-null) that was performed randomly. What is on the causal pathway between gene transfer and the cardiovascular phenotype cannot be established with certainty. Thus, we cannot state that the improvement of hepatic steatosis is causally linked with the prevention of heart failure. All effects must be obligatory caused by LDL receptor expression in parenchymal liver cells, which leads to a decrease of cholesterol levels. This decrease will induce a cascade of events in multiple organs. Changes in one organ may subsequently affect also other organs. However, this is a topic of speculation.

2) Consumption of HSHF diet made the mice to develop hepatic steatosis, mainly due to the increased dietary content in fructose. This may lead to elevations of LDL levels due to an inadequate clearance of these lipoproteins. The authors show that liver-specific overexpression of LDLr prevents HSHF diet-induced hepatic steatosis and leads to favorable evolution of cardiomyopathy in mice. Importantly, this does not occur in SC diet mice. This finding might suggest that the induction of hepatic steatosis as the key contributor of cardiomyopathy? Which might be the mechanism linking liver dysfunction and impaired myocardium? Is there any evidence showing such link in patients? If so, it was corrected upon cholesterol-lowering treatment (i.e., statins)? 

There is no direct or overwhelming evidence that NAFLD is independently associated with cardiomyopathy.

The following paragraph has been added in the Discussion:

‘Type 2 diabetes mellitus is a systemic disease that affects several organs. The HSHF diet model was characterized by the presence of hepatic steatosis, which is concordant with earlier observations[4].  Non-alcoholic fatty liver disease (NAFLD) is also highly prevalent in subjects with obesity and type 2 diabetes mellitus[1]. The global prevalence of NAFLD in the world is as high as one billion[2]. In patients with type 2 diabetes mellitus without known history of cardiac disease, NAFLD was associated with early features of left ventricular diastolic dysfunction[3].’

Statins have never been specifically tested to treat NAFLD and there are concerns that statins might induce liver toxicity in these patients.

REFERENCES

1.         Perumpail, B.J.; Khan, M.A.; Yoo, E.R.; Cholankeril, G.; Kim, D.; Ahmed, A. Clinical epidemiology and disease burden of nonalcoholic fatty liver disease. World J Gastroenterol 2017, 23, 8263-8276, doi:10.3748/wjg.v23.i47.8263.

2.         Loomba, R.; Sanyal, A.J. The global NAFLD epidemic. Nat Rev Gastroenterol Hepatol 2013, 10, 686-690, doi:10.1038/nrgastro.2013.171.

3.         Bonapace, S.; Perseghin, G.; Molon, G.; Canali, G.; Bertolini, L.; Zoppini, G.; Barbieri, E.; Targher, G. Nonalcoholic fatty liver disease is associated with left ventricular diastolic dysfunction in patients with type 2 diabetes. Diabetes Care 2012, 35, 389-395, doi:10.2337/dc11-1820.

4.         Ishimoto, T.; Lanaspa, M.A.; Rivard, C.J.; Roncal-Jimenez, C.A.; Orlicky, D.J.; Cicerchi, C.; McMahan, R.H.; Abdelmalek, M.F.; Rosen, H.R.; Jackman, M.R., et al. High-fat and high-sucrose (western) diet induces steatohepatitis that is dependent on fructokinase. Hepatology 2013, 58, 1632-1643, doi:10.1002/hep.26594.

Reviewer 2 Report

The paper improved very much. The authors well addressed my previous questions.

Author Response

The authors thank the reviewer for the positive analysis of our revised manuscript.

Round  3

Reviewer 1 Report

The authors answered to all my comments properly; however, the mice (fed the high fructose, high-fat diet) used in this study were no longer diabetic nor insulin resistant; therefore I recommended them to change their title, which in my opinion is particularly misleading. Authors should also check conclusions. The latter is probably due to the use of female instead of male mice, the former are more resistant to develop enhanaced adiposity and insulin resistance.

Another point, which I feel important, was that the HSHF diet mice rather developed hepatic steatosis as main diet complication. Notably, this may suggest a link between hepatic lipid accumulation, not diabetes or insulin resistance, on cardiac dysfunction. In my opinion, its mention and discussion should be at least added in a revised version of the discussion section of the ms. In this regard, are there studies(in either mice or humans) relating both complications?

FYI, I stated "accept with just minor revision" since my comments were just aimed at refocusing the discussion in light of the above-mentioned data.

I hope my comments have contributed to clarify my opinion on the revised version of Manuscript ID: ijms-474927.

Author Response

The authors have carefully read the new comments of the reviewer.

The authors answered to all my comments properly; however, the mice (fed the high fructose, high-fat diet) used in this study were no longer diabetic nor insulin resistant; therefore I recommended them to change their title, which in my opinion is particularly misleading. Authors should also check conclusions. The latter is probably due to the use of female instead of male mice, the former are more resistant to develop enhanaced adiposity and insulin resistance.

The authors respectfully disagree with the statement of the reviewer that these mice have no type 2 diabetes mellitus. As we have already indicated in our previous rebuttal letters, the data in Figure 3B show that the glucose levels in AAV8-null HSHF diet mice are significantly elevated compared to AAV8-SC diet mice. The absolute difference at 16 weeks is 1.72 mmol/L. Moreover, we state in lines 121-123 the following:

‘The HSHF diet induced hyperinsulinemia. Plasma insulin levels were 4.93-fold (p<0.001) and 3.98-fold (p<0.001) higher in AAV8-null HSHF diet mice and AAV8-LDLr HSHF diet mice, respectively, compared to respective SC diet groups (Figure 5A).’

Based on the presence of hyperinsulinemia and the presence of hyperglycemia, the conclusion that the HSHF diet induces insulin resistance and type 2 diabetes mellitus is completely justified.

The following sentence has been incorporated in the Discussion (lines 312-316) to highlight this conclusion, which is based on hard experimental results:

‘The HSHF diet resulted in marked hyperglycemia and marked hyperinsulinemia, indicating that insulin resistance and type 2 diabetes mellitus were induced by this diet. A limitation of the current study is that the weights of fat pads were not directly determined. However, the quantitative data on adipocyte hypertrophy provide evidence for the presence of enlargement of adipose tissue[1].’

Another point, which I feel important, was that the HSHF diet mice rather developed hepatic steatosis as main diet complication. Notably, this may suggest a link between hepatic lipid accumulation, not diabetes or insulin resistance, on cardiac dysfunction. In my opinion, its mention and discussion should be at least added in a revised version of the discussion section of the ms. In this regard, are there studies (in either mice or humans) relating both complications?

We discussed this point during the previous round. It is a well-known clinical fact that NAFLD is highly prevalent in subjects with obesity and type 2 diabetes mellitus. Several organs are affected in this disorder. This paper is focused on prevention of HFpEF. We referenced one paper that indicated that NALFLD might be associated with diastolic dysfunction. This does not prove causality since confounding should be considered.

The following text can be found in lines 316-323 of the revised manuscript:

‘Type 2 diabetes mellitus is a systemic disease that affects several organs. Therefore, multiple pathologies may cluster within the one and the same patient: HFpEF, diabetic nephropathy, and non-alcoholic fatty liver disease (NAFLD)[2]. The HSHF diet model was characterized by the presence of hepatic steatosis, which is concordant with earlier observations[3].  NAFLD is also highly prevalent in subjects with obesity and type 2 diabetes mellitus[4]. The global prevalence of NAFLD in the world is as high as one billion[5]. In patients with type 2 diabetes mellitus without known history of cardiac disease, NAFLD was associated with early features of left ventricular diastolic dysfunction[6].’

FYI, I stated "accept with just minor revision" since my comments were just aimed at refocusing the discussion in light of the above-mentioned data.

Taken together, during the previous round and this round, we have complied with the remarks of the reviewer. The authors are convinced that based on the experimental data, it is entirely justified that HSHF diet mice are characterized by type 2 diabetes mellitus and insulin resistance. Multiple pathologies may cluster in subjects with obesity and type 2 diabetes mellitus. This is also reflected by this model.

REFERENCES

1.         Jo, J.; Gavrilova, O.; Pack, S.; Jou, W.; Mullen, S.; Sumner, A.E.; Cushman, S.W.; Periwal, V. Hypertrophy and/or Hyperplasia: Dynamics of Adipose Tissue Growth. PLoS Comput Biol 2009, 5, e1000324, doi:10.1371/journal.pcbi.1000324.

2.         Targher, G.; Chonchol, M.; Bertolini, L.; Rodella, S.; Zenari, L.; Lippi, G.; Franchini, M.; Zoppini, G.; Muggeo, M. Increased risk of CKD among type 2 diabetics with nonalcoholic fatty liver disease. J Am Soc Nephrol 2008, 19, 1564-1570, doi:10.1681/ASN.2007101155.

3.         Ishimoto, T.; Lanaspa, M.A.; Rivard, C.J.; Roncal-Jimenez, C.A.; Orlicky, D.J.; Cicerchi, C.; McMahan, R.H.; Abdelmalek, M.F.; Rosen, H.R.; Jackman, M.R., et al. High-fat and high-sucrose (western) diet induces steatohepatitis that is dependent on fructokinase. Hepatology 2013, 58, 1632-1643, doi:10.1002/hep.26594.

4.         Perumpail, B.J.; Khan, M.A.; Yoo, E.R.; Cholankeril, G.; Kim, D.; Ahmed, A. Clinical epidemiology and disease burden of nonalcoholic fatty liver disease. World J Gastroenterol 2017, 23, 8263-8276, doi:10.3748/wjg.v23.i47.8263.

5.         Loomba, R.; Sanyal, A.J. The global NAFLD epidemic. Nat Rev Gastroenterol Hepatol 2013, 10, 686-690, doi:10.1038/nrgastro.2013.171.

6.         Bonapace, S.; Perseghin, G.; Molon, G.; Canali, G.; Bertolini, L.; Zoppini, G.; Barbieri, E.; Targher, G. Nonalcoholic fatty liver disease is associated with left ventricular diastolic dysfunction in patients with type 2 diabetes. Diabetes Care 2012, 35, 389-395, doi:10.2337/dc11-1820.